# Germline-somatic *JAK2* interactions are associated with clonal expansion in myelofibrosis

Derek W. Brown [1,2] ✉, Weiyin Zhou[1,3], Youjin Wang[1], Kristine Jones[1,3], Wen Luo[1,3], Casey Dagnall [1,3], Kedest Teshome[1,3], Alyssa Klein [1], Tongwu Zhang [1], Shu-Hong Lin[1], Olivia W. Lee[1], Sairah Khan [1], Jacqueline B. Vo[1,2], Amy Hutchinson[1,3], Jia Liu[1,3], Jiahui Wang[1,3], Bin Zhu[1,3], Belynda Hicks [1,3], Andrew St. Martin[4], Stephen R. Spellman [5], Tao Wang[4,6], H. Joachim Deeg[7], Vikas Gupta [8], Stephanie J. Lee[4,7], Neal D. Freedman[1], Meredith Yeager [1,3], Stephen J. Chanock [1], Sharon A. Savage [1], Wael Saber[4], Shahinaz M. Gadalla[1,9] & Mitchell J. Machiela [1,9] ✉

Myelofibrosis is a rare myeloproliferative neoplasm (MPN) with high risk for progression to acute myeloid leukemia. Our integrated genomic analysis of up to 933 myelofibrosis cases identifies 6 germline susceptibility loci, 4 of which overlap with previously identified MPN loci. Virtual karyotyping identifies high frequencies of mosaic chromosomal alterations (mCAs), with enrichment at myelofibrosis GWAS susceptibility loci and recurrently somatically mutated MPN genes (*e.g.*, *JAK2*). We replicate prior MPN associations showing germline variation at the 9p24.1 risk haplotype confers elevated risk of acquiring *JAK2*$^{V617F}$ mutations, demonstrating with long-read sequencing that this relationship occurs in *cis*. We also describe recurrent 9p24.1 large mCAs that selectively retained *JAK2*$^{V617F}$ mutations. Germline variation associated with longer telomeres is associated with increased myelofibrosis risk. Myelofibrosis cases with high-frequency *JAK2* mCAs have marked reductions in measured telomere length – suggesting a relationship between telomere biology and myelofibrosis clonal expansion. Our results advance understanding of the germline-somatic interaction at *JAK2* and implicate mCAs involving *JAK2* as strong promoters of clonal expansion of those mutated clones.

Myelofibrosis (MF) is a rare myeloproliferative neoplasm (MPN), with an incidence of ~1 per 100,000 per year[1,2], characterized by the development of abnormal hematopoietic stem cell (HSC) clones and altered bone marrow microenvironment, leading to fibrosis[3,4]. Patients with MF typically develop cytopenia due to proliferation of aberrant HSC clones and hepatosplenomegaly due to extramedullary hematopoiesis[4]. Individuals with MF are at a high risk of developing acute myeloid leukemia[5]. MF can present as primary disease (primary MF) or progress from another MPN (secondary MF) such as polycythemia vera or essential thrombocythemia[6].

Recurrent somatic driver mutations have been identified in MF, particularly in *JAK2*, *MPL*, and *CALR*[7–9]; current evidence also indicates a heritable component as well[10,11]. So far, the *JAK2* 46/1 haplotype and a single nucleotide polymorphism (SNP) in the *TERT* gene region have been established as predisposition alleles for MPNs, including both primary and secondary MF[12–16], which have led to the use of *JAK*

inhibitors and telomerase inhibitors as potential therapeutic agents for MF[17–19]. A genome-wide association study (GWAS) of MPN that included 136 MF patients identified additional MPN-associated loci (e.g., 3q21.3, 3q25.33, 6p21.31, 13q14.11, 18q11.2, 21q22.12)[20]. Current knowledge of MF genetic etiology is inferred from the MPN data.

In this work, we undertook an integrated approach to investigate the genetics of MF which includes analysis of germline variation, somatic *JAK2* point mutations, somatic mosaic chromosomal alterations (mCAs), and leukocyte telomere length among MF patients who underwent hematopoietic cell transplantation (HCT) and reported to the Center for International Blood and Marrow Transplant Research (CIBMTR), representing a clinically important subset of MF patients. We report here six MF susceptibility loci, four of which replicate prior MPN findings at 9p24.1 (*JAK2*), 5p15.33 (*TERT*), 3q25.33 (*IFT80*), and 4q24 (*TET2*). We show germline variation at the 9p24.1 risk haplotype confers elevated risk of acquiring *cis JAK2^{V617F}* mutations which are selectively retained by recurrent 9p24.1 mCAs. MF is dynamically associated with telomere length in which longer inherited telomere length increases MF risk, and clonal expansion of 9p24.1 mCAs markedly reduces measured telomere length.

## Results

### CIBMTR MF case characteristics

In total, 937 MF cases met the criteria for inclusion in our study, the majority of which were male (58.06%) and had DNA collected at an average age of 56.9 years (median = 58.4, IQR = 52.3–63.9; Supplemental Table 1). Most cases were primary (68.84%) MF, with intermediate 1 or 2 disease (49.30%) based on the Dynamic International Prognostic Scoring System (DIPSS)[21] score. The average time from diagnosis to transplant in the full cohort was 63.5 months (median = 25.1, IQR = 9.0–88.3). Compared to primary MF cases, secondary MF cases were more likely to be female (54.45% vs. 36.28%) and had longer average time from diagnosis to transplant (122.8 months vs. 36.8 months).

### MF susceptibility loci identified with estimated large effect size

We performed a GWAS using 827 MF cases and 4135 ancestry-matched cancer-free controls drawn from the Prostate, Lung, Colorectal, and Ovarian (PLCO) Screening Trial[22] (Methods section). The liability scale heritability of MF was estimated to be 11.4% (s.e.= 5.8%). The genomic inflation ($\lambda$) and intercept from linkage disequilibrium score regression (LDSC)[23] showed minimal evidence for systematic inflation ($\lambda = 1.02$, LDSC intercept= 1.01, Fig. 1).

We analyzed 9,672,066 genotyped and imputed germline variants after filtering on control minor allele frequency (>0.5%) and imputation quality score (>0.7; Methods section) and identified six independent genome-wide significant loci ($P < 5 \times 10^{-8}$) (Supplemental Table 2 and Fig. 1), four of which replicated prior MPN findings at 9p24.1 (*JAK2*), 5p15.33 (*TERT*), 3q25.33 (*IFT80*), and 4q24 (*TET2*)[20]. We observed rs7851556 in 9p24.1 as the most significant variant (odds ratio (OR) = 2.39, 95% confidence interval (CI) = 2.13–2.68, $P = 5.75 \times 10^{-51}$), which is in strong linkage disequilibrium (LD) with the lead variant reported for MPN, rs1327494 ($R^2_{EUR} = 0.95$, $D'_{EUR} = 0.98$)[20] and located in an intron of *JAK2*, a gene which promotes cellular proliferation through the JAK/STAT pathway[8,24]. As chromosomal alterations are common at 9p24.1 (see mCA section below), we performed a sensitivity analysis in individuals with no detectable 9p24.1 mCAs to ensure no miscalling of germline variants resulting from mCAs in the 9p24.1 region; the rs7851556 signal remained significant (OR = 1.65, 95% CI = 1.41–1.92, $P = 1.99 \times 10^{-10}$), although the effect estimate was attenuated due to removal of MF cases that carried the *JAK2* risk haplotype (see below section on *JAK2*–mCA relationship). rs7705526 had the strongest association (OR = 1.65, 95% CI = 1.48–1.84, $P = 7.62 \times 10^{-19}$) in the 5p15.33 locus. Previously identified in MPN[20], this intronic variant is located in *TERT*, which encodes telomerase, the reverse transcriptase that extends telomeric DNA repeats, and has been associated with CD34+ to CD45+ ratio[25]. The 3q25.33 variant rs201009932 (OR = 5.78, 95% CI = 3.67–9.11, $P = 4.06 \times 10^{-14}$) (Supplemental Table 2) is in moderate LD with the MPN variant rs77249081 ($R^2_{EUR} = 0.25$, $D'_{EUR} = 1.00$)[20] and resides in an intron variant of *IFT80*, a part of the IFT complex essential for the assembly and maintenance of cilia as well as differentiation through the Sonic Hedgehog pathway[26,27]. At 4q24, rs1548483, located near *TET2*, a putative tumor suppressor gene and common somatic driver mutation in clonal hematopoiesis[28,29], was significant (OR = 2.27, 95% CI = 1.71-3.01, $P = 1.42 \times 10^{-8}$); notably, this variant is in high LD with the previously identified MPN variant rs62329718 ($R^2_{EUR} = 0.94$, $D'_{EUR} = 1.00$)[20]. Conditional analyses controlling for the lead GWAS variant for the 9p24.1 and 5p15.33 loci identified no evidence for additional independent signals (Supplemental Fig. 1).

Our GWAS identified two additional MF germline susceptibility loci: 6p21.32 and 17p13.1 (Supplemental Table 2 and Fig. 1). The 6p21.32

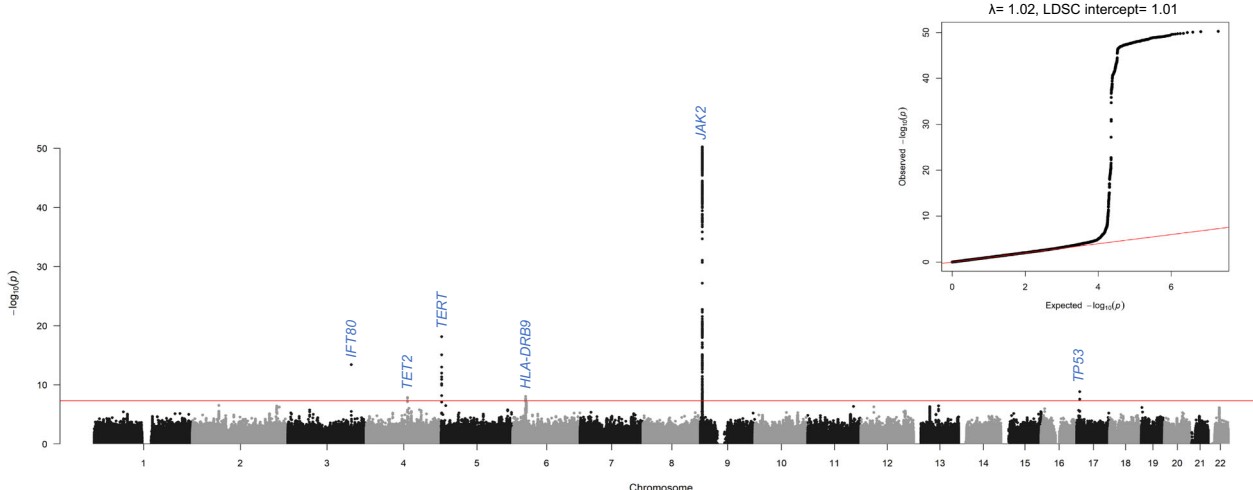

**Fig. 1 | Manhattan plot and quantile-quantile plot from the myelofibrosis genome-wide association study.** The association -log₁₀ *P*-values are plotted for each tested genetic variant on the *y*-axis (two-sided, logistic regression) and chromosomal position on the x-axis. The nearest gene for each identified locus is labeled. The red line indicates the genome-wide significance threshold ($5 \times 10^{-8}$). The quantile–quantile plot displays observed versus expected −log₁₀ *P*-values. Source data are provided as a Source Data file.

variant, rs28442287, (OR = 1.63, 95% CI = 1.38–1.93, $P = 9.34 \times 10^{-9}$) is a downstream variant of *HLA-DRB9*, within the human leukocyte antigen system[30,31], and has been implicated in bone marrow related diseases[32,33]. The 17p13.1 variant, rs78378222, (OR = 4.47, 95% CI = 2.75–7.27, $P = 1.56 \times 10^{-9}$) is a 3′ UTR variant of the commonly mutated tumor suppressor gene *TP53* that increases susceptibility to different types of cancer (e.g., skin basal cell carcinoma, prostate cancer, glioma, and lymphocytic leukemia)[34–37]. Future studies are warranted to validate these two MF germline susceptibility loci.

We performed stratified GWAS investigating primary MF (569 cases, 2845 controls) and secondary MF (258 cases, 1290 controls) to investigate possible differences (Supplemental Table 2), and no informative differences in association signals were observed between the two MF types (Supplemental Table 2, Supplemental Fig. 2). We further stratified secondary MF analyses into post-polycythemia vera MF (119 cases, 595 controls) and post-essential thrombocythemia MF (139 cases, 695 controls; Supplemental Table 3). We observed a strong genome-wide significant signal at 9p24.1 (*JAK2*) only in post-polycythemia vera MF (Supplemental Table 3 and Supplemental Fig. 3), which is consistent with prior reports of higher *JAK2* involvement in patients with polycythemia vera[38–40].

## MF-associated germline variants near *JAK2* increase expression

We performed expression quantitative trait locus (eQTL) analyses using genome-tissue expression (GTEx) whole blood data[41] and identified eQTLs with *JAK2* expression (e.g., rs7847141; OR = 1.05, 95% CI = 1.03-1.08, $P = 3.69 \times 10^{-5}$). Germline variants at the lead 9p24.1 locus additionally colocalized with *JAK2* expression with rs7851556 having the highest Posterior Probability (PP; PP = 0.59; Supplemental Fig. 4)[42]. MF risk alleles were associated with increased levels of *JAK2* expression, explaining 35% of the shared association signal at 9p24.1 ($PP_{SNP} = 0.35$)[42]. Colocalization was also detected at 6p21.32 with *HLA-DRB9* (colocalization PP = 0.36)[43].

In analysis of whole blood GTEx expression data, a transcriptome-wide association study (TWAS)[41,44] identified *JAK2* ($Z = 9.00$, $P = 2.18 \times 10^{-19}$) and *RP11-39K24.4* ($Z = 10.78$, $P = 4.26 \times 10^{-27}$) as significant genes ($P < 3.59 \times 10^{-6}$; Supplemental Figure 5), with positive *JAK2* expression associated with increased MF risk. Conditional analyses were performed with predicted *RP11-39K24.4* expression, and *JAK2* remained an independent expression signal (rs7851556 conditional GWAS $P = 9.90 \times 10^{-26}$; Supplemental Fig. 6).

## *JAK2* germline risk haplotype confers elevated risk of *cis JAK2^V617F* mutations

Targeted PacBio long-read sequencing of the *JAK2* region identified 562 (60.82%) individuals with the commonly observed activating *JAK2^V617F* mutation, a known MPN driver mutation[7,8]. Secondary MF following a prior diagnosis of polycythemia vera or essential thrombocythemia were more likely to have the *JAK2^V617F* mutation than primary MF cases (67.93% vs. 57.57%, $P = 2.85 \times 10^{-3}$), with post-polycythemia vera MF having the highest frequency of the *JAK2^V617F* mutation (97.04% vs. 57.57%, $P = 5.82 \times 10^{-10}$). The estimated average mutation allelic fraction on background haplotypes was 62.54% (median = 69.76, IQR = 37.58-92.74), suggesting high clonal expansion of *JAK2^V617F* mutated clones. MF cases carrying the risk allele (T) of rs7851556 (our lead GWAS SNP) were more likely to acquire a somatic *JAK2^V617F* mutation ($P = 9.41 \times 10^{-14}$) (Supplemental Table 4). Additionally, individuals with the *JAK2* 46/1 germline risk haplotype (GGC, from rs3780367, rs10974944, rs12343867, Methods section) were substantially more likely to acquire *JAK2^V617F* mutations (OR = 2.69, 95% CI = 2.02-3.58, $P = 1.16 \times 10^{-11}$), with 634 (68.61%) individuals carrying the *JAK2* 46/1 germline risk haplotype, and post-polycythemia vera MF (88.15%) having a higher frequency than both primary MF (65.98%) and post-essential thrombocythemia MF (66.45%). Furthermore, when examining phase information, we observed a strong *cis* relationship

between the germline risk haplotype and *JAK2^V617F* mutations acquired on the same risk haplotype (binomial $P = 1.23 \times 10^{-26}$; Supplemental Table 5), as previously observed in MPN patients[12–14]. These results were consistently observed when stratified by type of MF: primary MF (binomial $P = 1.86 \times 10^{-13}$), post-polycythemia vera MF (binomial $P = 4.88 \times 10^{-10}$), and post-essential thrombocythemia MF (binomial $P = 1.90 \times 10^{-4}$). Of the 562 individuals carrying a *JAK2^V617F* mutation, 370 (65.84%) had the mutation in *cis* with the risk haplotype (binomial $P = 7.68 \times 10^{-33}$). Interestingly, during our *JAK2^V617F* mutation calling (Methods section), we identified 5 individuals with evidence of the somatic mutation potentially acquired independently on both germline haplotypes (Supplemental Table 6) which were replicated in independent sequencing runs on new libraries. Future studies are needed to further explore the frequency of independent *JAK2^V617F* mutations on both germline haplotypes in MF cases.

## Chromosomal alterations are abundant in MF and preferentially expand *JAK2^V617F* clones

At least one detectable autosomal mCA was detected in 684 (73.31%) individuals which is in contrast to ~3% in population-based surveys[45,46]. An elevated frequency of mCAs in secondary MF compared to primary MF was also observed (78.35% vs. 71.03%, OR = 1.48, 95% CI = 1.06-2.05, $P = 0.0196$), with post-polycythemia vera MF having the highest frequency of mCAs (94.12% vs. 71.03%, OR = 6.53, 95% CI = 3.13-13.60, $P = 5.54 \times 10^{-7}$). Recurrent copy neutral loss of heterozygosity (CNLOH) events were detected on chromosome 9p ($N = 298$; 31.94%), and recurrent loss events were observed on chromosome 13q ($N = 89$; 9.54%) and 20q ($N = 92$; 9.86%; Fig. 2). Each GWAS susceptibility locus showed enrichment for mCAs compared to age and sex-matched cancer-free individuals in the UK Biobank[45,46] (binomial $P < 1 \times 10^{-8}$; Supplemental Table 7), suggesting mCAs could clonally expand MF risk conferring alleles at susceptibility loci.

Since the *JAK2* 9p24.1 locus was the most notable in our GWAS and contains a hotspot of *JAK2^V617F* mutations, we closely examined mCAs in this region. In total, 378 (40.51%) individuals had a detectable autosomal mCA across the 9p24.1 region. MF cases carrying the risk allele (T) of rs7851556 (our lead GWAS SNP) were more likely to have mCAs within the 9p24.1 region ($P = 2.28 \times 10^{-19}$), with CNLOH representing over 75% of the observed mCAs ($N = 294$; Supplemental Table 8). These results replicate previous reports of an association between 9p CNLOH and the *JAK2* 46/1 risk haplotype[45]. We performed allelic shift analyses in heterozygous individuals to test for a *cis* relationship between the *JAK2* risk haplotype and mCAs spanning 9p24.1[45,46]. Using highly correlated genotyped proxy SNPs to our lead GWAS SNP rs7851556[47], we found that the *JAK2* risk haplotype is predominantly amplified by gains (23 of 28, binomial $P = 9.12 \times 10^{-4}$, proxy SNP rs10815167), retained by losses (14 of 16, binomial $P = 4.18 \times 10^{-3}$, proxy SNP rs2230724), and duplicated by CNLOH (51 of 59, binomial $P = 9.05 \times 10^{-9}$, proxy SNP rs10815167), providing strong evidence for preferential clonal expansion of mCAs with the *JAK2* MF risk haplotype.

Using long-read PacBio sequencing data, we noted substantial allelic imbalance of heterozygous variants in the vicinity of *JAK2* in individuals with mCAs spanning the region, providing independent confirmation of mCA calls in the region (Supplemental Fig. 7). Of the 374 individuals with detectable mCAs spanning *JAK2* who also had long-read sequencing data, 97.86% had a *JAK2^V617F* mutation, indicating a strong relationship between mCAs and *JAK2^V617F* mutations ($P = 4.37 \times 10^{-80}$; Supplemental Table 9). To investigate the clonal evolutionary history of *JAK2* mutations, we examined mutated cellular fractions of *JAK2^V617F* and *JAK2* mCAs. We found substantially higher *JAK2^V617F* allelic fractions on background haplotypes compared to estimated mCA cellular fractions, suggesting the acquisition of a *JAK2^V617F* mutation occurred prior to acquiring a *JAK2* chromosomal alteration for the vast majority of MF cases (binomial $P = 3.30 \times 10^{-54}$; Supplemental Fig. 8). The estimated *JAK2^V617F* allelic fractions and mCA

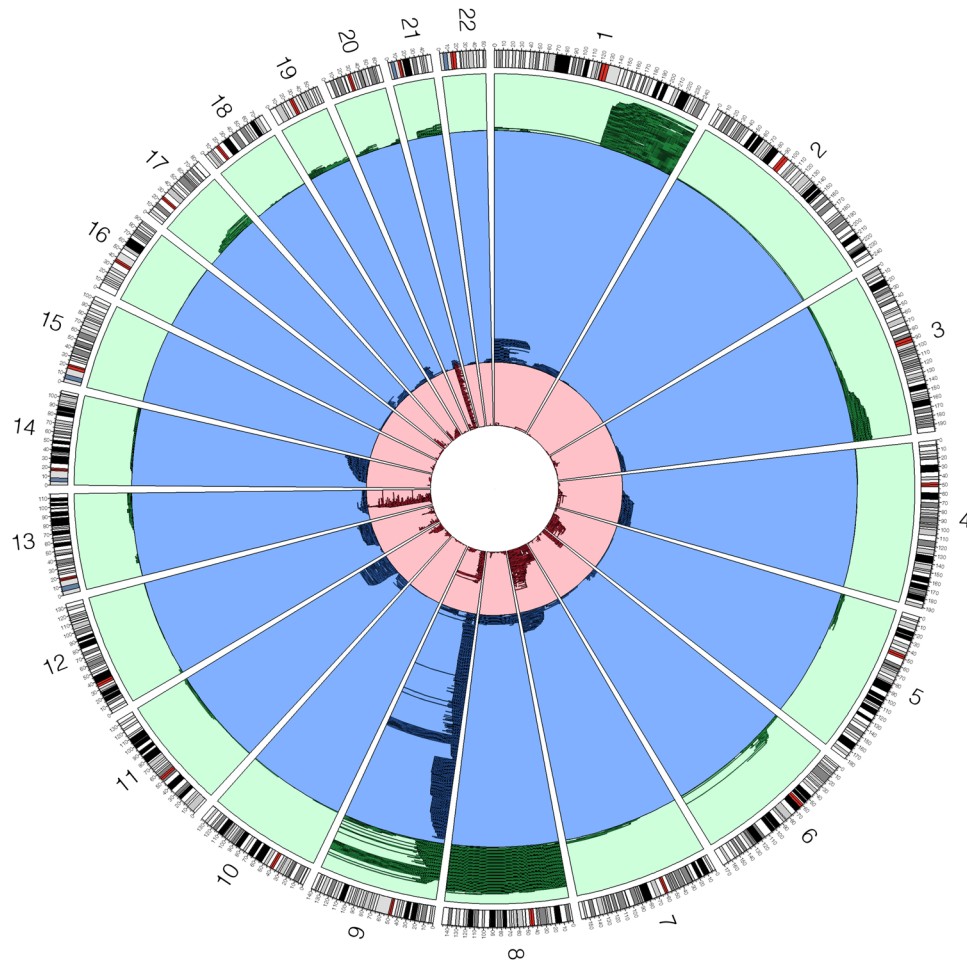

**Fig. 2 | Mosaic chromosomal alterations by autosomal location.** Green events indicate mosaic copy gains, blue events indicate copy neutral loss of heterozygosity, and red events indicate mosaic losses. The highlighted region on each chromosome band indicates the centromere location. Source data are provided as a Source Data file.

cellular fractions may be lower than the true somatic fraction in the actual diseased myeloid cells because we used whole blood DNA from most of the patients. However, this would not affect the ratio of *JAK2*^V617F allelic fraction to mCA cellular fraction. mCAs were also enriched across *MPL* and *CALR* gene positions (binomial $P < 5 \times 10^{-32}$; Supplemental Table 10), suggesting mCAs may also clonally expand other MPN driver mutations similar to the *JAK2* locus, although this hypothesis needs to be further studied.

### Inherited longer telomere length associated with MF risk

In addition to the *JAK2* MF susceptibility region, the 5p15.33 locus near *TERT* implicates telomere length in MF risk. To evaluate the role of telomere length, we used a panel of germline variants associated with measured telomere length to develop a polygenic risk score (PRS) for inherited telomere length[48] and observed a strong positive association between increased genetically-inferred leukocyte telomere length and increased MF risk (OR = 1.33, 95% CI = 1.23-1.44, $P = 2.56 \times 10^{-13}$). Of the 19 telomere-length associated variants imputed, seven (rs4691895, rs7705526, rs2853677, rs228595, rs62053580, rs75691080, and rs34978822) were nominally associated ($P < 0.05$) with MF risk (binomial $P = 2.31 \times 10^{-5}$). The allele related to longer telomere length was associated with increased risk of MF for five of these seven variants (Fig. 3 and Supplemental Table 11). The telomere length PRS was associated with the presence of *JAK2*^V617F mutations (OR = 1.20, 95% CI = 1.04–1.37, $P = 0.01$) as well as mCAs (OR = 1.17, 95% CI = 1.01–1.36, $P = 0.04$), suggesting longer telomere length may afford cells the

ability to clonally expand to detectable clonal fractions, after acquiring somatic mutations.

In a Mendelian randomization (MR) analysis to evaluate a directional relationship between the telomere length-associated variants and MF risk, the intercept from MR-Egger regression was non-significant ($P = 0.65$, Table 1) after removing five potentially pleiotropic variants (including the lead *TERT* variant identified in our GWAS) (Supplemental Table 11)[49], suggesting no pleiotropy[50]. Each MR method utilized indicated a strong increasing effect between the telomere length genetic instrument and MF risk (Table 1 and Supplemental Fig. 9).

The genetic correlation between leukocyte telomere length and MF using LD score regression[51] was estimated based on summary statistics from a published telomere length GWAS[48] along with summary statistics from our MF GWAS. A marginally significant genome-wide genetic correlation was observed between telomere length and MF (LDSC $r = 0.23$, s.e.m. = 0.11, $P = 0.038$), similar in magnitude to what was reported for telomere length and MPN (LDSC $r = 0.19$, s.e.m. = 0.09, $P = 0.037$)[20]. Together, our results indicate longer telomere length is associated with increased risk of MF when evaluated by established leukocyte telomere length variants and genome-wide.

### Telomere length attrition and mCA-induced clonal expansion

In an analysis of leukocyte relative telomere length (rTL) measured before HCT, on average, telomere length was significantly shorter among older subjects ($P = 0.0037$), peripheral blood mononuclear cell

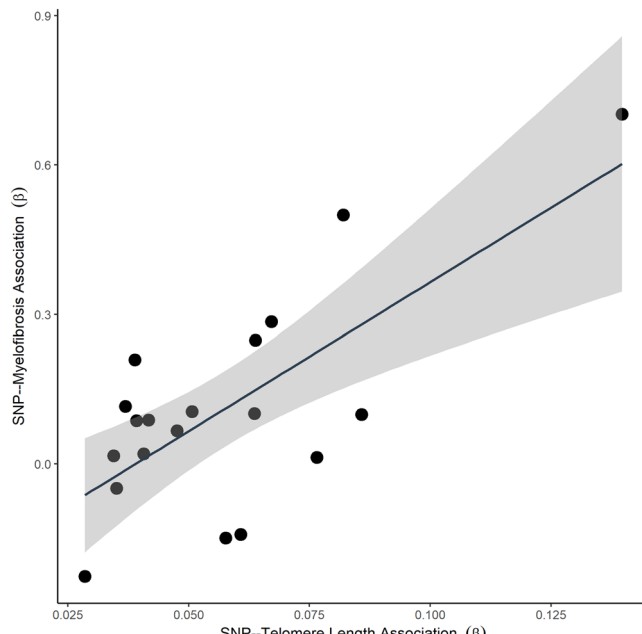

**Fig. 3 | The effect of each variant on genetically-inferred telomere length and myelofibrosis risk.** Estimates for the SNP-telomere length (Li et al.) and SNP-myelofibrosis associations are presented in Supplemental Table 11. A linear model estimated trend line and calculated 95% confidence interval around the trend (gray fill) are plotted (two-sided $P = 5.48 \times 10^{-4}$).

(PBMC)-derived DNA ($P = 0.0031$), secondary MF ($P = 0.0016$), and those with a longer time from diagnosis to HCT transplant ($P = 0.0171$) (Supplemental Table 12). To further investigate the reported association between telomere length and mCAs in the general population[52], individuals with autosomal mCAs had significantly shorter rTL compared to individuals without autosomal mCAs ($P = 7.93 \times 10^{-5}$; Supplemental Table 13). Multivariable analyses demonstrated negative associations between rTL and the presence of any autosomal mCA (OR = 0.14, 95% CI = 0.04–0.44, $P = 7.39 \times 10^{-4}$; Table 2). These results were consistent when restricted to individuals with whole blood-derived DNA (OR = 0.19, 95% CI = 0.06–0.62, $P = 6.05 \times 10^{-3}$). A close inspection of mCAs in the 9p24.1 region demonstrated stronger attenuation of rTL (OR = 0.04, 95% CI = 0.01–0.19, $P = 2.60 \times 10^{-5}$), indicating individuals with autosomal mCAs, especially those spanning *JAK2*, exhibit signatures of clonal expansion. Multivariable Poisson regression also demonstrated rTL was negatively associated with a greater number of autosomal mCAs (incidence rate ratio = 0.33, 95% CI = 0.22–0.48, $P = 1.98 \times 10^{-8}$). Associations between measured telomere length and mCAs were consistent when stratified by

### Table 1 | Mendelian randomization results using variants and summary statistics from Li et al.[a]

| Method | OR (95% CI) | p-value[b] |
|---|---|---|
| Maximum-likelihood | 8.04 (3.41, 18.93) | $1.88 \times 10^{-6}$ |
| Simple median | 6.25 (2.04, 19.10) | $1.32 \times 10^{-3}$ |
| Weighted median | 5.07 (1.75, 14.70) | $2.76 \times 10^{-3}$ |
| IVW[c] | 7.69 (3.29, 17.97) | $2.43 \times 10^{-6}$ |
| MR-Egger | 13.49 (1.07, 169.93) | $4.41 \times 10^{-2}$ |
| Intercept | 0.97 (0.85, 1.11) | 0.6475 |

[a]Five variants (rs4691895, rs7705526, rs34991172, rs228595, rs34978822) were detected to have evidence of pleiotropy (FDR < 0.2) as detailed in Supplemental Table 11 and removed from the MR analyses.
[b]All reported tests are two-sided.
[c]Inverse-variance weighted.

chromosomal location (e.g., telomeric or interstitial) ($P_{\text{het}} = 0.91$) and copy number state (e.g., gain, loss, CNLOH, undetermined; $P_{\text{het}} = 0.90$, Table 2). In sensitivity analyses, adjusting for MF type, DIPSS score, and time to transplant, negative associations with rTL were observed both overall and stratified by chromosomal region and copy number state (overall OR = 0.19, 95% CI = 0.06–0.64, $P = 7.44 \times 10^{-3}$; Table 2).

Analyses of the rTL and mCA clonal fractions indicated a strong relationship in which increasing mCA cellular fraction was associated with a substantial decrease in measured rTL ($\beta = -0.57$, 95% CI = -0.74 to −0.39, $P = 4.76 \times 10^{-10}$), especially for mCAs spanning *JAK2* ($\beta = -1.17$, 95% CI = -1.44 to −0.91, $P = 1.39 \times 10^{-16}$). Interestingly, we did not observe an association between measured rTL and $JAK2^{V617F}$ mutation presence (OR = 0.84, 95% CI = 0.29-2.42, $P = 0.74$), but we did observe an association with $JAK2^{V617F}$ clonal fraction ($\beta = -1.11$, 95% CI = −1.31 to −0.91, $P = 2.26 \times 10^{-25}$). After removing individuals with mCAs, rTL was no longer associated with $JAK2^{V617F}$ clonal fraction ($\beta = -0.27$, 95% CI = -0.71–0.16, $P = 0.22$), suggesting $JAK2^{V617F}$ alone may not be a strong driver of rapid clonal expansion in MF (Supplemental Table 14). These findings indicate mCAs, many of which selectively retain or duplicate $JAK2^{V617F}$ mutations, promote rapid clonal expansion of mutated clones resulting in significant reductions in telomere length.

## Discussion

We conducted an integrated genomic characterization of MF by investigating both germline susceptibility alleles and somatic events, particularly mosaic chromosomal events. Our study supports a key role of *JAK2* events as a critical driver of MF and underscores important interactions of germline susceptibility with $JAK2^{V617F}$ mutations and mCAs[12–14,45]. We observed that 68% of MF cases carry at least one germline *JAK2* risk allele, whereas somatic events are also critical to MF development - in our study 61% of MF cases carried a somatic $JAK2^{V617F}$ mutation, and 41% with an mCA spanning *JAK2*. Overall, we observed that ~85% of cases involve *JAK2*. Using independent sequencing runs, we identified five individuals with evidence of acquiring independent $JAK2^{V617F}$ mutations on both germline haplotypes. The frequency and consequences of independent $JAK2^{V617F}$ mutations on both germline haplotypes should be further studied. In a GWAS, we identified six MF susceptibility loci (one including *JAK2* on 9p24.1) with two independent signals unique to MF[20]. The estimated MF heritability is 11.4% (s.e. = 5.8%), and all MF susceptibility loci had high effect sizes (OR > 1.6) relative to those typically found by GWAS, suggesting a strong germline component at multiple genomic loci for MF risk. Global assessment of mCAs demonstrated a high frequency in MF cases and demonstrated enrichment at each GWAS locus and across other MPN driver mutations, providing potential evidence for genome-wide germline-somatic interactions beyond that observed for *JAK2*.

Our integrated study demonstrates a complex germline-somatic interaction in MF patients at the 9p24.1 susceptibility locus, which confirmed prior findings and revealed insights into MF etiology. The observed effect of telomere length on disease risk in concert with 9p24.1 indicates that the presence of a germline susceptibility locus involving *JAK2* directly influences the probability of developing a somatic event in the same region, presumably on the same haplotype (Fig. 4). We observed that individuals with the germline *JAK2* risk haplotype tagged by rs7851556 were predisposed to acquiring a somatic $JAK2^{V617F}$ mutation in *cis*, as previously reported[12–14]. The *cis* relationship could not be checked directly due to the distance between our lead GWAS variant (rs7851556) and the $JAK2^{V617F}$ mutation, but this relationship is supported by the high LD between rs7851556 and variants (rs3780367, rs10974944, rs12343867) in the 46/1 risk haplotype ($R^2 > 0.93$). We also observed mCAs lead to preferential over-representation of this risk haplotype containing the $JAK2^{V617F}$ mutation[45]. Prior studies demonstrate *JAK2* is a strong activator of cellular growth and proliferation[8,24], promotes cell surface localization[53], is activated in response to a variety of cytokines[54–56], and

**Table 2 | Association between measured telomere length and autosomal mCA status by chromosomal region and copy number change**

| | Univariable model | | Multivariable model[a] | | | Multivariable model[b] | | |
|---|---|---|---|---|---|---|---|---|
| | OR (95% CI) | Association p-value[c] | OR (95% CI) | Association p-value[c] | Het p-value[d] | OR (95% CI) | Association p-value[c] | Het p-value[d] |
| Overall | 0.12 (0.04–0.34) | $8.31 \times 10^{-5}$ | 0.14 (0.04–0.44) | $7.39 \times 10^{-4}$ | – | 0.19 (0.06–0.64) | $7.44 \times 10^{-3}$ | – |
| Chromosomal region | – | – | – | – | 0.9057 | – | – | 0.9241 |
| Telomeric | 0.08 (0.02–0.25) | $1.72 \times 10^{-5}$ | 0.11 (0.03–0.39) | $6.06 \times 10^{-4}$ | – | 0.16 (0.04–0.59) | $5.98 \times 10^{-3}$ | – |
| Interstitial | 0.07 (0.02–0.26) | $3.99 \times 10^{-5}$ | 0.10 (0.03–0.37) | $4.98 \times 10^{-4}$ | – | 0.14 (0.04–0.56) | $5.34 \times 10^{-3}$ | – |
| Copy number change | – | – | – | – | 0.9032 | – | – | 0.7922 |
| Gain | 0.08 (0.02–0.33) | $4.24 \times 10^{-4}$ | 0.10 (0.02–0.47) | $3.11 \times 10^{-3}$ | – | 0.13 (0.03–0.65) | 0.0132 | – |
| Loss | 0.06 (0.02–0.23) | $2.08 \times 10^{-5}$ | 0.06 (0.02–0.24) | $5.39 \times 10^{-5}$ | – | 0.09 (0.02–0.38) | $1.05 \times 10^{-3}$ | – |
| CNLOH | 0.05 (0.02–0.18) | $2.83 \times 10^{-6}$ | 0.08 (0.02–0.28) | $1.29 \times 10^{-4}$ | – | 0.11 (0.03–0.44) | $1.91 \times 10^{-3}$ | – |
| Undetermined | 0.04 (0.002–0.62) | 0.0220 | 0.03 (0.001–0.69) | 0.0279 | – | 0.02 (0.001–0.60) | 0.0243 | – |

*CNLOH* copy neutral loss of heterozygosity; *mCA* mosaic chromosomal alteration.
[a]Multivariable models control for sex, age, age-squared, genetic ancestry, and DNA source.
[b]Multivariable models additionally control for myelofibrosis type, DIPSS score, and time to transplant.
[c]Two-sided logistic regression.
[d]Denotes test of heterogeneity of effect within subgroups analyzed.

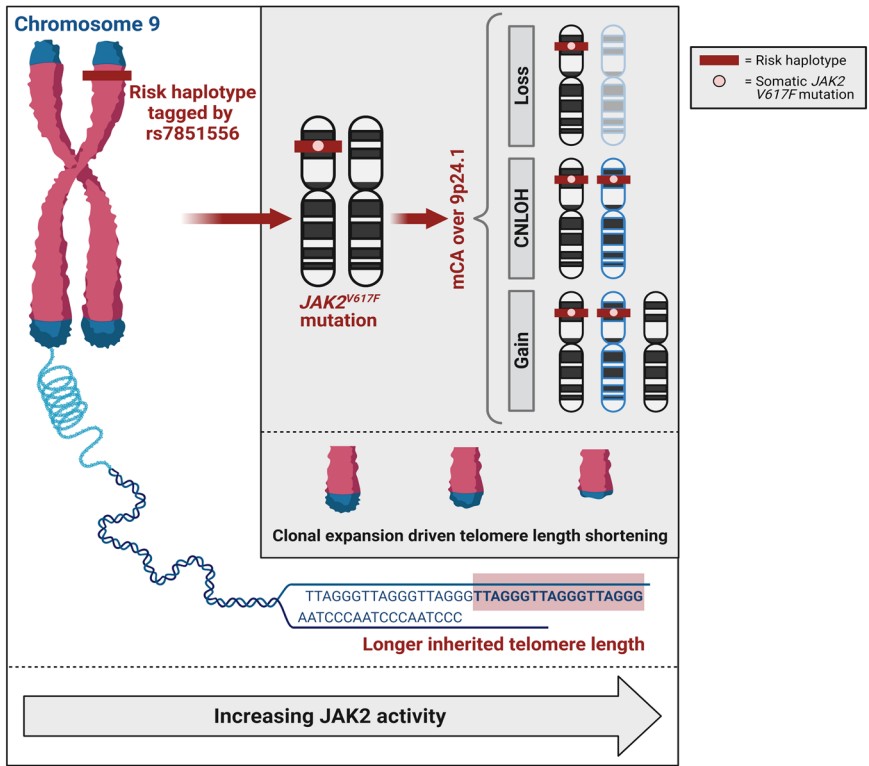

**Fig. 4 | A proposed conceptual framework of myelofibrosis genetic etiology.** Both longer inherited telomere length and the germline *JAK2* risk haplotype tagged by rs7851556 are associated with increased risk of myelofibrosis. The germline *JAK2* risk haplotype further predisposes to somatically acquire the *JAK2^V617F* mutation in *cis*, and mosaic chromosomal alterations in the 9p24.1 region lead to over- representation of the germline risk haplotype and *JAK2^V617F* somatic mutation. We hypothesize that increasing JAK2 activity underlies each of these processes which leads to increased clonal expansion and proliferation, and accelerated telomere length shortening in myelofibrosis patients.

induces bone marrow fibrosis[57]. Altered JAK2 activity conferred by the germline *JAK2* risk haplotype, *JAK2^V617F* activating mutation, and 9p24.1 mCAs could lead to a cellular phenotype characterized by increased clonal expansion. While we were able to connect germline suscept- ibility and somatic mutations in cross-sectional data from MF patients, future longitudinal assessment will be key in follow-up of our study.

Likewise future studies characterizing germline functional variation are needed to better understand how the germline *JAK2* MF suscept- ibility locus leads to altered *JAK2* expression and acquisition of *JAK2^V617F* mutations.

The MF susceptibility signal near *TERT* together with our PRS, MR, and genetic correlation analyses, suggest that polygenetic effects of

genetic variants associated with telomere length are important contributors to MF predisposition, especially in light that telomere length is required for hematopoietic stem cell self-renewal[58]. A prior study on a smaller number of variants suggested longer genetically determined telomere length, as derived by telomere length associated variants, increases MPN risk[59]. Our observations further suggest inherited genome-wide variation promoting longer telomeres could fuel MF clonal expansion, especially *JAK2*-mCA mediated expansion. A downstream consequence of this increased cellular proliferation is shorter telomeres as observed in the reduced rTL in patients with mCAs. As previously demonstrated in MPN patients, and replicated in our MF study, measured rTL was not associated with *JAK2*[V617F] mutation presence, but was inversely associated with *JAK2*[V617F] clonal fraction[60]. Our results indicate that mCAs are driving this observed association between measured rTL and *JAK2*[V617F] clonal fraction, suggesting mCAs, particularly mCAs spanning *JAK2*, promote clonal expansion in MF. A recent trial identified that telomerase inhibitor therapy may be an effective treatment for MF; although the mechanism is poorly understood[19]. It is possible that patients with MF and *JAK2* mCAs who have expanded clones may benefit from telomerase inhibition through senescence or apoptosis of these MF clones, potentially leading to better clinical outcomes[61]. Future functional and clinical investigations, using the flow-FISH assay to measure leukocyte cell-type specific telomere length, are required to test this hypothesis. Furthermore, our study included MF patients who received HCT between 2000 and 2016, and may represent a specialized clinical subset, which may not generalize to all MF patients.

The large number of cases for this rare disease allowed for investigation of potential differences between primary and secondary MF, and secondary subtype. We noted no major differences in germline susceptibility by type of MF, and observed a significant signal at 9p24.1 (*JAK2*) in post-polycythemia vera MF. We did observe differences in their somatic profiles, with both *JAK2*[V617F] mutations and mCA acquisition more commonly observed in secondary MF compared to primary MF. This observation underscores a higher somatic load of mutations in secondary MF, which could be due to prior MPN diagnosis and longer disease duration, and is in agreement with prior reports[38–40]. This higher frequency of somatic mutations could have implications for MF treatment approaches, with *JAK2* inhibitors potentially inducing a greater response in individuals with higher mutation allele burden[62].

In conclusion, our integrated genomic investigation of MF highlights a strong inherited genetic component of MF, which, in turn, influences the somatic profile of abundant mCAs in MF. These findings underscore the critical role of *JAK2* and telomere biology in MF susceptibility, which could inform avenues for improved clinical management of MF.

## Methods
### Study population
This study utilized blood samples and clinical information from the Center for International Blood and Marrow Transplant Research (CIBMTR) database and repository (https://www.cibmtr.org). The study was approved by the National Marrow Donor Program institutional review board. All patients provided written informed consent for the research use of their samples and clinical data. Patients eligible for inclusion were those who underwent hematopoietic cell transplantation (HCT) for primary idiopathic or secondary MF between 2000 and 2016 and have a pre-HCT blood sample available for genomic analysis. Blood samples, from either whole blood or peripheral blood mononuclear cells (PBMCs), were collected within 30 days before administering HCT conditioning regimen. In total, 937 MF patients met the criteria for inclusion in our study, of which 863 (92.1%) contributed whole blood and 74 (7.9%) PBMCs (Supplemental Table 1).

### SNP genotyping and quality control assessment
We used Qiagen QIAsymphony for DNA extraction. All genomic laboratory work included in this study was conducted by the NCI Cancer Genomics Research Laboratory. Genotyping on MF cases was completed using the Illumina Infinium Global Screening Array-24v1-0. Genotypes were called using standard Illumina microarray data analysis workflows. Controls were selected from 62,880 previously genotyped cancer-free individuals within the Prostate, Lung, Colorectal, and Ovarian (PLCO) Screening Trial[22], who were genotyped on the same array as the cases. To minimize technical artifacts, genotyping quality control steps were performed on the joint set of MF cases and PLCO controls. Standard quality control checks were performed to ensure high completion rates (≥95%), no sample contamination, sex concordance, no unexpected duplicates or replicates, normal rates of heterozygosity, and no instances of high relatedness (IBD < 0.2). Genetic ancestry was inferred using SNPWEIGHTS[63], which estimates the percentage of European, West African, and East Asian ancestry for each subject. After filtering based on quality control steps and European ancestry (>80%), 833 MF cases and 56,929 cancer-free controls were eligible for the GWAS analysis.

### Genome-wide association study
We genetically matched the MF cases and PLCO controls using PCA-matchR to minimize the effects of confounding due to potential population stratification bias[64]. For the combined set of cases and controls, we extracted linkage disequilibrium filtered variants ($R^2 < 0.1$) from the array manifest file (GSA-24v2-0_A1) and performed principal component (PC) analyses using PLINK[65]. The first 20 PCs and eigenvalues were used to 1:5 match MF cases and PLCO controls based on eigenvalue weights (Supplemental Figure 10). After matching, six MF cases did not have well matching controls and were removed from the analysis (Supplemental Figure 10), resulting in a total of 827 cases and 4135 genetically matched controls. Imputation was carried out using the Michigan Imputation Server with the TOPMed reference panel (https://imputationserver.sph.umich.edu). Following imputation, association analyses were conducted under an additive model using SNPTEST (https://mathgen.stats.ox.ac.uk/genetics_software/snptest/snptest.html). Any PCs that remained significant after matching were additionally adjusted for in the GWAS association tests. For the main GWAS, germline variants were filtered based on control minor allele frequency (>0.5%) and imputation quality score (>0.7). In GWAS stratified by MF type (Supplemental Table 1), variants were filtered based on a more stringent control minor allele frequency (>5%) to remove potential spurious associations arising due to small sample size. Manhattan plots for results visualization were generated using the "qqman" and "hudson" R packages[66,67].

### Linkage disequilibrium score regression
Linkage disequilibrium score regression (LDSC) was used to estimate the narrow-sense heritability estimate of MF risk as well as compute genetic correlations[51]. We computed LD scores from European individuals within 1000 Genomes Project data[68]. Variants were filtered by minor allele frequency (>1%); 5,956,602 variants were retained for all calculations[51]. Heritability calculations were based on a MF sample prevalence of 0.1667 in our GWAS and a population prevalence of $5.69 \times 10^{-5}$[51].

### Colocalization of MF GWAS and QTLs
Colocalization analyses of the MF GWAS signals were performed using expression quantitative trait locus (eQTL) whole blood genome-tissue expression (GTEx) (version 8) data[41] with eCAVIAR and Hyprcoloc[42,43]. We extracted GWAS and QTL summary statistics 100 kb upstream and downstream of the lead GWAS SNP as input for colocalization analyses, with the exception of chromosome 6, which was limited to 10 kb, due to the number of variants in the region. For each MF GWAS locus,

eCAVIAR colocalization posterior probability (colocalization $PP > 0.01$)[43] or Hyprcoloc posterior probability ($PP > 0.50$)[42] was used to identify colocalization.

## Transcriptome-wide association study

We performed a transcriptome-wide association study (TWAS) on the MF GWAS summary statistics using FUSION (http://gusevlab.org/projects/fusion/)[44]. TWAS was performed using GTEx whole blood gene expression data (version 7)[41] by imputing the gene expression phenotypes for GWAS data overlapping the LD reference panel (1000 Genomes European)[51]. FUSION was used to test associations using pre-computed expression reference weights. A Bonferroni corrected level of significance of $3.59 \times 10^{-6}$ (0.05/13,909 GTEx available features) was used to assess statistical significance.

## Targeted PacBio sequencing and mutation calling

Targeted PacBio Single Molecule Real-Time (SMRT) sequencing was performed on MF cases to detect $JAK2^{V617F}$ mutations. We followed the SMRT sequencing process protocols as described in the PacBio sequencing manual (https://www.pacb.com/wp-content/uploads/Procedure-checklist-Preparing-SMRTbell-libraries-using-PacBio-barcoded-M13-primers-for-multiplex-SMRT-sequencing.pdf). Briefly, a 2-step PCR, using 10 ng of DNA as input, was conducted. In step 1, template-specific primers amplified the region of chr9:5067166-5074380 in GRCh38. Next, in step 2, unique barcode sequences were incorporated onto each sample, using universal tags, for multiplexing. Products were purified and quantified with an additional normalization step to ensure equal concentration of each sample prior to pooling, and hairpin adapters were ligated to the ends of each amplicon pool (up to 384 samples) during the SMRT bell library preparation. Each pooled library had primer annealing and polymerase binding performed according to the protocol and sequenced on 1 Sequel II SMRT Cell 8 M. After sequencing, Circular Consensus Sequencing (CCS) read generation was performed with set criteria, including a minimum of 3 passes and accuracy of 99%. CCS reads were used as input to lima (https://github.com/PacificBiosciences/barcoding) to demultiplex the pooled samples, and then aligned to GRCh38 using pbmm (https://github.com/PacificBiosciences/pbmm2).

Haplotypes were then generated based on aligned CCS reads, including 3 SNPs within the $JAK2$ 46/1 haplotype (rs3780367, rs10974944, rs12343867) and the $JAK2^{V617F}$ mutation (chr9:5073770). For each MF case, any haplotype with frequency <1% was removed. Additionally, cases were removed if they did not have sufficient DNA quantity ($N = 1$), total number of CCS reads < 1000 ($N = 1$), more than two germline haplotypes (based on rs3780367, rs10974944, rs12343867; $N = 4$), or $JAK2^{V617F}$ mutations called on more than one germline haplotype ($N = 49$). We repeated sequencing of 53 individuals that failed QC; 1 individual did not have sufficient DNA for resequencing. We also resequenced an additional 43 MF cases who passed the above QC steps to further validate our PacBio sequencing and QC procedure. We followed the same protocol as detailed above for the resequenicng effort, with new aliquots of the genomic DNA taken and reamplified to generate a new library. Again, haplotypes were generated based on aligned CCS reads, including the 3 SNPS within the $JAK2$ 46/1 haplotype and the $JAK2^{V617F}$ mutation, and any haplotype with frequency <1% was removed. We observed a high degree of concordance (>93%) between the first and second sequencing efforts for the subjects who originally passed our QC procedure ($N = 43$). Of those individuals that originally failed our sequencing QC ($N = 53$), we removed cases that were again identified with more than two germline haplotypes ($N = 2$) or $JAK2^{V617F}$ mutations called on more than one germline haplotype ($N = 5$). Overall, 924 MF cases were retained for downstream analyses with average read depth of 8,623.2 (median = 6872, min = 3155, max = 27,402).

## Mosaic chromosomal alterations QC and calling

Using genotype data, MF cases were called for mCAs. Before mCA calling, the same GWAS quality control measures were performed, with completion rates relaxed to ≥90%. MoChA software (https://github.com/freeseek/mocha) was used to detect somatic copy number aberrations, a similar approach has been previously implemented[45,46]. Briefly, MoChA utilizes hidden Markov models (HMM) to integrate B allele frequency (BAF) and $\log_2$ R ratio (LRR), and leverage haplotype information to detect subtle imbalances between maternal and paternal allelic fractions in a cell population. The BAF was calculated as the ratio of signal intensity between two alleles at each genotyped variant in relation to estimated genotype clusters and was used to detect allelic imbalances as well as calculate the proportion of cells with a deletion, duplication, and copy neutral loss of heterozygosity (CNLOH)[69]. Contiguous genomic stretches of BAF values for heterozygous SNPs that deviate from 0.5 are indicative of mosaic chromosomal alterations. LRR calculates the log base 2 of the ratio of observed total signal intensities to expected signal intensities for a genotyped variant[69]. Contiguous genomic stretches with LRR > 0 indicate copy gain, <0 indicate loss and around 0 indicate CNLOH. Furthermore, phase data was used to detect subtle over or under representation of haplotypes indicating the presence of a mCA. Eagle2, a software utilizing a population-based approach to infer phase (1000 Genomes reference panel), was used to infer haplotypes[70,71].

Chromosome 9 had a high frequency of CNLOH events with high cell fractions. These events are poorly detected by phase-based methods, due to lack of heterozygous sites in the event region. Although MoChA applies a non-phased-based model to detect high-level mosaic events, this approach only detects mosaic gains or losses, not CNLOH. Therefore, we additionally applied a custom software pipeline that utilized the BAFSegmentation software (http://baseplugins.thep.lu.se/wiki/se.lu.onk.BAFsegmentation) to recover additional events not detected in MoChA[72]. Chromosome 9 was segmented for mosaic events using circular binary segmentation on BAF values. All potential events from both detection methods were plotted and visualized, and false positive calls were excluded from the analysis based on manual review of each plot.

Samples with called mCAs on the merged set from MoChA and BAFSegmentation were classified by copy number state (gain, loss, CNLOH, or undetermined events), cellular fraction (the percentage of sampled leukocytes carrying the detected mCA), and chromosomal region (e.g., telomeric, interstitial, or whole chromosome event). Events that only occurred around telomeric ends (±1 Mb from chromosome ends) were defined as telomeric, events that spanned an entire chromosome were defined as whole chromosome events, and all other events were defined as interstitial.

## Leukocyte telomere length polygenetic risk score and Mendelian randomization

We generated a telomere length polygenic risk score (PRS) by aggregating variants previously found to be associated with measured telomere length in GWAS into a weighted genetic instrument (Supplemental Table 11)[48]. The inherited telomere length PRS was standardized to have mean 0 and standard deviation 1. PRS analyses were adjusted for ancestry principal components within MF risk analyses, and sex, age, age-squared, genetic ancestry, and DNA source within $JAK2^{V617F}$ mutation and mCA analyses. Mendelian randomization (MR) analyses were performed using the telomere length associated variants[48] within the "Mendelian Randomization" R package[50]. We utilized the "GLIDE" R package to investigate any potential evidence of pleiotropy among the included variants[49]. Any variant found to have heterogeneous effects between measured telomere length and MF (false discovery rate <0.2) was removed from the analysis[49].

## Telomere length measurement

We utilized a modified qPCR assay to measure leukocyte relative telomere length (rTL) in pre-HCT extracted DNA[73]. The qPCR assay calculates the ratio between telomeric repeat copy number (T) and that of a single reference gene (beta-globin gene; 36B4) (S). Relative T/S was calculated in relation to a reference curve and final measurements were exponentiated to assure normality. All telomeric and 36B4 reactions were measured in triplicate, and the mean was used for final calculations. A total of 927 patients had DNA available for telomere length measurement. The overall completion rate was high (98.9%), resulting in 916 patients with available measured telomere length. An internal control, which is used to standardize results within the project, had an overall coefficient of variation (CV) of 2.96% and the intraclass correlation coefficient (ICC) and its 95% confidence interval for study technical replicates was 0.982 (0.975, 0.986). All telomere length analyses adjusted for sex, age, age-squared, genetic ancestry, and DNA source, unless otherwise noted.

## Reporting summary

Further information on research design is available in the Nature Research Reporting Summary linked to this article.

## Data availability

Raw SNP genotyping data and raw targeted PacBio sequencing data that were generated on this study from the myelofibrosis individuals is available on dbGaP under accession number phs002635.v1.p1. CIBMTR supports accessibility of research in accord with the National Institutes of Health (NIH) Data Sharing Policy and the National Cancer Institute (NCI) Cancer Moonshot Public Access and Data Sharing Policy. The CIBMTR only releases de-identified datasets that comply with all relevant global regulations regarding privacy and confidentiality. The mCA calls and phenotypic UK Biobank data used in this study, which were used under license, are available from: http://www.ukbiobank.ac.uk/. Genotype data from the Prostate, Lung, Colorectal, and Ovarian (PLCO) Screening Trial is available on dbGaP under accession number phs001286.v2.p2. Source data are provided with this paper.

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

## Acknowledgements

This work was supported by the intramural research program of the Division of Cancer Epidemiology and Genetics, National Cancer Insti-tute, and National Institutes of Health. The Cancer Genomics Research Laboratory is funded with Federal funds from the National Cancer Institute, National Institutes of Health, under NCI Contract No.

75N910D00024. The CIBMTR is supported primarily by Public Health Service U24CA076518 from the National Cancer Institute (NCI), the National Heart, Lung and Blood Institute (NHLBI) and the National Institute of Allergy and Infectious Diseases (NIAID); HHSH250201700006C from the Health Resources and Services Administration (HRSA); and N00014-20-1–2705 and N00014-20-1-2832 from the Office of Naval Research; Support is also provided by Be the Match Foundation, the Medical College of Wisconsin, the National Marrow Donor Program. This work used the computational resources of the NIH's High-Performance Computing Biowulf cluster and was conducted using the UK Biobank resource (application 21552). The UK Biobank was established by the Wellcome Trust, the Medical Research Council, the United Kingdom Department of Health, and the Scottish Government. The UK Biobank has also received funding from the Welsh Assembly Government, the British Heart Foundation, and Diabetes UK. The conceptual framework of myelofibrosis pathogenesis was created with BioRender.com. We thank Dr. Ian D. Buller for his statistical support and guidance. The opinions expressed by the authors are their own and this material should not be interpreted as representing the official viewpoint of the U.S. Department of Health and Human Services, the National Institutes of Health or the National Cancer Institute.

## Author contributions

D.W.B., Y.W., S.M.G., and M.J.M. conceived the study. A.S.M., S.R.S., T.W., S.J.L., and W.S. contributed samples. K.J., W.L., C.D., and K.T. performed the experiments. W.Z. carried out the mCA calls. D.W.B., W.Z., A.K., T.Z., and O.W.L. performed computational and statistical analyses. J.L., J.W., B.Z., and N.D.F. contributed to genetic analysis of PLCO. H.J.D., and V.G. provided subject matter expertize. S.M.G. and M.J.M. jointly supervised the study. D.W.B., S.J.C., S.A.S., S.M.G., and M.J.M. drafted the manuscript with input from all authors. D.W.B., W.Z., Y.W., K.J., W.L., C.D., K.T., A.K., T.Z., S.-H.L., O.W.L., S.K., J.B.V., A.H., J.L., J.W., B.Z., B.H., A.S.M., S.R.S., T.W., H.J.D., V.G., S.J.L., N.D.F., M.Y., S.J.C., S.A.S., W.S., S.M.G., and M.J.M. critically read and approved the final version of the manuscript.

## Funding

## Competing interests

The authors declare no competing interests.

## Additional information

[1]Division of Cancer Epidemiology and Genetics, National Cancer Institute, Rockville, MD, USA. [2]Cancer Prevention Fellowship Program, Division of Cancer Prevention, National Cancer Institute, Rockville, MD, USA. [3]Cancer Genomics Research Laboratory, Frederick National Laboratory, Frederick, MD, USA. [4]Center for International Blood and Marrow Transplant Research, Medical College of Wisconsin, Milwaukee, WI, USA. [5]Center for International Blood and Marrow Transplant Research, National Marrow Donor Program, Minneapolis, MN, USA. [6]Division of Biostatistics, Medical College of Wisconsin, Milwaukee, WI, USA. [7]Clinical Research Division, Fred Hutchinson Cancer Research Center, Seattle, WA, USA. [8]MPN Program, Princess Margaret Cancer Centre, University of Toronto, Toronto, ON, Canada. [9]These authors jointly supervised this work: Shahinaz M. Gadalla, Mitchell J. Machiela. ✉e-mail: derek.brown@nih.gov; mitchell.machiela@nih.gov

