## [Peer review file · Nature Communications]

REVIEWER COMMENTS

Reviewer #1 (Remarks to the Author): Expert in blood cancer genetics

Polycythemia vera (PV), essential thrombocythemia (ET) and myelofibrosis (MF) constitute the group of myeloproliferative disorders (MPN). These disorders are grouped together is that the differential diagnosis between them is often not clear-cut, and that MF often evolves from PV or ET. As a result, these disorders are expected to show, and are known to shown, a genetic overlap, both at the germline and somatic level.

Brown et al report a GWAS on 827 MF cases vs 4135 controls. They detect six loci at $P < 5 \times 10^{-8}$, four of which are known (JAK2, TERT, IFT80, and TET2). Further, they detect two borderline-significant signals (at HLA and TP53), and an enrichment of low MF GWAS P-values among variants previously reported to associate with telomere length. The paper is well written, but the findings are not novel.

The identified JAK2 lead variant rs7851556 is in perfect LD with the well-known JAK2 MPN risk variant, which has been reported by several authors since 2009 and is known to predispose for acquisition of somatic JAK2 V617F mutation in cis on the same chromosome (PMID 19287385, PMID19287384, PMID19287382 and PMID33057200). The higher expression of JAK2 in blood is also known.

The TERT, IFT80 and TET2 variants are also known MPN variants. The pleiotropy between MPN and telomere length has been demonstrated previously (PMID33057200), and the identified TERT variant was recently reported to influence blood CD34+ stem cell levels (PMID 35007327).

The HLA and TP53 associations are borderline-significant (order of $P = 10^{-9}$; thus not Bonferroni-significant), and are not subjected to replication analysis. The latter two should therefore be reported with caution, if at all.

In conclusion, this study mainly replicates things that are already known. The fact that the authors underplay this in the abstract and the text is annoying and, to my mind, almost dishonest. The authors do have a point in that their cohort comprises a larger number of MF cases than previous MPN studies, which are dominated by PV and ET. Accordingly, the way forward seems to be to rewrite the paper such that it clearly states what it is already known, and what is novel, including the precise relationship between their findings and previously published (including for example precise LD statistics between identified and previously reported lead variants). The authors also need to remove all kinds of premature speculation, for example “our results indicate clonal expansion of JAK2 is facilitated by germline variation associated with longer telomere length” and “The findings have translation

implications, highlighting the potential role for telomerase inhibitors as treatment in high-risk individuals”.

Reviewer #2 (Remarks to the Author): Expert in MF and MPN molecular genetics and genomics

The authors provide a comprehensive genomic analysis on very a large number of myelofibrosis patients compared with an even 5 times larger number of genetically-matched, cancer-free individuals serving as controls.

The main methods comprise i) SNP genotyping to identify germline susceptibility loci, ii) calling for mosaic chromosomal alterations to determine copy number state, cellular fraction, and chromosomal region of the respective events, and iii) qPCR-based measurement of the relative telomere leukocyte length for association of identified SNPs with myelofibrosis risk and proliferation.

The findings are in line with previous data on this topic and extend our understanding of the etiology of myelofibrosis due to new insights how germline variants and somatic alterations interact regarding dysregulation of JAK2 activity and thus clonal expansion in myelofibrosis patients.

In addition to known susceptibility loci in JAK2, TERT, TET2, and IFT80, the authors identified two novel germline loci in the genes HLA-DRB9 and TP53 predisposing for myelofibrosis acquisition. Furthermore, the autosomal location of both recurrent CNLOH events (mainly on chromosome 9p) and copy-number alterations (mainly recurrent loss events on chromosome 20q and 13q) is valid compared with former SNP array studies in myeloproliferative neoplasm patients.

Including their data on telomere length, the authors finally provide a model in which the germline JAK2 risk haplotype predisposes to somatic acquisition of JAK2 V617F, subsequently causing clonal expansion and cell proliferation, and finally leading to accelerated telomere length shortening in myelofibrosis patients. These data and the proposed model justify the title of the manuscript.

In summary, the presented manuscript is methodologically sound, very elaborate, thoughtfully written, and provides significant novelty to the field as well as for the readership of Nature Communications.

Minor remarks:

- The study is actually based on data from 827 myelofibrosis patients after quality control and matching with cancer-free individuals, not on 937 cases as stated in the Abstract.
- The paper is in general a bit tough to read due to the high number of very particular abbreviations that are used. Please check for readability.
- Dynamic International Prognostic Scoring System = DIPSS (not DIPPS)

Reviewer #3 (Remarks to the Author): Expert in MF and MPN genomics

In the present work authors perform an integrated genomic analysis on a wide cohort of primary and secondary myelofibrosis (PMF and SMF) patients in order to characterize the heritable genetic component of these neoplasms. As a result, authors identify six novel germline susceptibility loci, and among them two variants emerged, the first in 9p24.1 locus within an intron of the frequently mutated JAK2 gene, and the second located within an intron of TERT gene. Authors show a correlation between the previously described JAK2 46/1 risk haplotype and the occurrence of JAK2V617F mutation or mosaic chromosomal alterations involving 9p24.1. According to their hypothesis, the presence of both germline variants, together with the acquisition of JAK2V617F mutation and somatic mosaic chromosomal alterations, contributes to the increased JAK2 activity and clonal expansion as demonstrated by the reduction of telomere length. The presented results are significant and original, but I have some concerns about data interpretation.

Major points

1. Myelofibrosis (MF) belongs to chronic Philadelphia-Negative Myeloproliferative Neoplasms (MPNs); it is generally assumed that the malignant transformation involves the myeloid lineage while lymphoid cells do not belong to the neoplastic clone, indeed in most cases these cells do not harbor the driver mutation observed in patients. Authors performed their analysis by studying whole blood specimens or PBMNCs therefore including in their analysis both neoplastic myeloid cells belonging to the expanded malignant clone (granulocytes, monocytes and circulating CD34+cells) and lymphoid cells probably representing a non-malignant cell fraction. Does the presence of both clonal and non-clonal cells in the analyzed sample might influence analysis results? Can the variable frequency of neoplastic cells in the analyzed sample influence analysis results? In such a complex and variable scenario how were germline variants distinguished from somatic ones?
2. The present study is focused on MF that can be either primary or secondary, when it develops in patients who have a previous diagnosis of polycythemia vera (PV) or essential thrombocythemia (ET). Even if these two conditions (PMF and SMF) have been considered very similar for a long time, recent studies highlighted several differences in both clinical features and molecular characteristics. Related to the clinics, specific prognostic models were developed for risk prediction in SMF patients, in particular MYSEC-PM (Myelofibrosis Secondary to PV and ET – Prognostic Model). Taking this into account, it would be important to include, in Supplemental Table 2, the frequency of driver mutations (affecting JAK2, MPL ad CALR) harbored by patients included in the study. Moreover, it is crucial to specify the frequency of Post-PV MF and Post-ET MF and not only the frequency of SMF. Finally, I warmly suggest to consider MYSEC-PM prognostic classification for SMF patients.
3. Related to the type of SMF, in Supplementary Table 2 would it be possible to perform the analysis by distinguishing between PostPV-MF and PostET-MF?

4. In paragraph “JAK2 germline risk haplotype confers elevated risk of cis JAK2V617F mutations” authors evaluate the association between a previously described JAK2 germline risk haplotype (JAK2 46/1) and the presence of JAK2V617F mutation. The authors report that the frequency of JAK2V617F in the study cohort is about 60% and is more frequent in SMF patients compared to PMF. This is obviously expected because it is already known that almost the totality of PV patients (therefore PostPV-MF patients) harbor JAK2V617F driver mutation. Which is the frequency of the JAK2 46/1 risk haplotype in the study cohort? Since the frequency of JAK2V617F mutation is greater in PostPV-MF patients compared to PMF and PostET-MF ones, would it be possible to evaluate the correlation between the presence of this mutation and the germline risk haplotype separately for each disease?

5. The authors demonstrate that JAK2V617F is frequently observed in cis with the JAK2 46/1 risk haplotype. Please can the authors provide any observation about the co-occurrence of JAK2V617F mutation and the rs7851556 variant? Patients carrying the rs7851556 are more likely to have JAK2V617F mutation?

6. JAK2V617F was the first driver mutation discovered in 2005 in MPN patients. It is the most frequent driver mutation affecting the vast majority (95%) of PV patients and almost 60% of PMF and ET ones. In the present study authors report that the interaction between germline and somatic variants in JAK2 contributes to the clonal expansion observed in MF patients. Can the authors discuss the possible involvement of the same mechanisms in other MPNs (PV and ET)? Can it be generalized that the presence of the risk haplotype rs7851556 is associated with an increased risk of acquiring a JAK2V617F mutation rather than the development of myelofibrosis?

7. Authors says that the presence of JAK2 risk haplotype, JAK2V617F mutation and 9p24.1 mosaic chromosomal alterations confer increased JAK2 activity. Can the authors provide any evidence or discussion on how germline haplotype might be able to influence protein activity, since the novel identified risk variant and those previously described are intronic variants that are not located in promoter region?

8. Authors report that relative telomer length is inversely correlated with the increased JAK2V617F clonal fraction. As said by the authors, JAK2V617F mutated patients represent only about 60% of patients included in their study cohort. Can telomer length be correlated with clonal expansion even in cases where the MPN driver mutation is represented by CALR or MPL variants?

Minor points

- Line 94: Authors say that primary myelofibrosis (PMF) patients “had an average time from diagnosis to transplant of 63.5 months (median= 25.1, IQR=9.0-88. 3).”, then, in line 96 it is reported that secondary myelofibrosis (SMF) “had longer average time from diagnosis to transplant (122.8 months vs. 36.8 months)”, it is not clear why the two mean values reported for primary myelofibrosis are different.

Author response

We thank the Reviewers for their time and insightful comments. We have crafted a revision that we hope will be acceptable to the Reviewers.

This is an issue we would like to comment on that slightly alters the number of *JAK2* mutations, identified in deeper QC analyses due to the past comments of our Reviewers. During our original targeted PacBio sequencing and *JAK2*^{V617F} mutation calling, we removed 54 individuals in the QC process based on low sequencing depth (total CCS reads < 1000; N= 1), having more than two germline haplotypes (N= 4), or having *JAK2*^{V617F} mutations called on more than one germline haplotype (N= 49). We resequenced 54 individuals to more thoroughly interrogate *JAK2*^{V617F} mutations within our myelofibrosis cases. One individual did not have sufficient DNA for resequencing, so this resequencing effort was conducted on **53 individuals** that previously failed QC. Of the 53 individuals that originally failed our sequencing QC:

- N= 5 were identified to have the *JAK2*^{V617F} mutation on two germline haplotypes
- N= 2 were identified to have 3 germline haplotypes
- N= 46 passed all QC steps and were added to our downstream analyses

We retained the 46 individuals that passed the QC steps in the downstream mutation analyses and removed the above 7 individuals who had *JAK2*^{V617F} mutations on multiple haplotypes or had >2 germline haplotypes (*i.e.*, potential evidence for sample contamination).

Our new mutation analyses now include a total of 924 (878 + 46) myelofibrosis subjects with the *JAK2*^{V617F} mutation identified in 562 (60.8%) individuals. We have updated the **Results**, **Online Methods**, and relevant tables and figures to reflect this work, and there were no substantive changes in the overall conclusions of our manuscript based on these updates.

Interestingly, we identified 5 individuals with the *JAK2*^{V617F} mutation on two germline haplotypes in both sequencing runs. This may be evidence that these individuals acquired two independent *JAK2*^{V617F} mutations. We have created a table of these 5 individuals (**Supplemental Table 6**) which details their germline haplotypes, and number of reads that carry the *JAK2*^{V617F} mutation. These results have been added to the manuscript as well, but out of caution these 5 individuals were not included in the downstream mutation analyses. Within the **Results** section: “Interestingly, during our *JAK2*^{V617F} mutation calling (**Online Methods**), we identified 5 individuals with evidence of the somatic mutation potentially acquired independently on both germline haplotypes (**Supplemental Table 6**) which were replicated in independent sequencing runs on new libraries. Future studies are needed to further explore the frequency of independent *JAK2*^{V617F} mutations on both germline haplotypes in MF cases.” Within the **Discussion** section: “Using independent sequencing runs, we identified 5 individuals with evidence of acquiring independent *JAK2*^{V617F} mutations on both germline haplotypes. The frequency and consequences of independent *JAK2*^{V617F} mutations on both germline haplotypes should be further studied.”

Below, please find our responses to each of the Reviewer comments, as well as where the corresponding revisions can be found. All changes have been tracked with **highlighted text**.

Supplemental Table 6. Identified individuals with *JAK2*^{V617F} mutations potentially acquired on both germline haplotypes

Subject ID ^a	Germline Haplotype	Original Sequencing		Resequencing	
		Haplotype Reads	JAK2 ^{V617F} Mutation Count ^b	Haplotype Reads	JAK2 ^{V617F} Mutation Count ^b
1	GGC	4,067	128 (3.15)	10,683	353 (3.30)
1	TCT	5,223	2,318 (44.38)	14,424	5,991 (41.53)
2	GGC	183	77 (42.08)	1,572	1,005 (63.93)
2	TCT	5,484	2,803 (51.11)	25,398	12,342 (48.59)
3	GGC	379	135 (35.62)	1,653	683 (41.32)
3	TCT	10,698	157 (1.47)	28,923	693 (2.40)
4	GGC	3,286	226 (6.88)	9,479	804 (8.48)
4	TCT	5,086	4,194 (82.46)	14,691	12,197 (83.02)
5	GGC	3,368	167 (4.96)	10,619	254 (2.39)
5	TCT	3,076	199 (6.47)	11,274	843 (7.48)

^aSubject ID 2 was diagnosed with post-polycythemia vera myelofibrosis, all other subjects were diagnosed with primary myelofibrosis

^bPercentage of germline haplotype reads carrying the mutation

Reviewer #1 (Remarks to the Author): Expert in blood cancer genetics

Polycythemia vera (PV), essential thrombocythemia (ET) and myelofibrosis (MF) constitute the group of myeloproliferative disorders (MPN). These disorders are grouped together is that the differential diagnosis between them is often not clear-cut, and that MF often evolves from PV or ET. As a result, these disorders are expected to show, and are known to shown, a genetic overlap, both at the germline and somatic level.

Brown et al report a GWAS on 827 MF cases vs 4135 controls. They detect six loci at $P < 5 \times 10^{-8}$, four of which are known (JAK2, TERT, IFT80, and TET2). Further, they detect two borderline-significant signals (at HLA and TP53), and an enrichment of low MF GWAS P-values among variants previously reported to associate with telomere length. The paper is well written, but the findings are not novel.

The identified JAK2 lead variant rs7851556 is in perfect LD with the well-known JAK2 MPN risk variant, which has been reported by several authors since 2009 and is known to predispose for acquisition of somatic JAK2 V617F mutation in cis on the same chromosome (PMID 19287385, PMID19287384, PMID19287382 and PMID33057200). The higher expression of JAK2 in blood is also known.

The TERT, IFT80 and TET2 variants are also known MPN variants. The pleiotropy between MPN and telomere length has been demonstrated previously (PMID33057200), and the identified TERT variant was recently reported to influence blood CD34+ stem cell levels (PMID 35007327).

The HLA and TP53 associations are borderline-significant (order of $P = 10^{-9}$; thus not Bonferroni-significant), and are not subjected to replication analysis. The latter two should therefore be reported with caution, if at all.

In conclusion, this study mainly replicates things that are already known. The fact that the authors underplay this in the abstract and the text is annoying and, to my mind, almost dishonest. The authors do have a point in that their cohort comprises a larger number of MF cases than previous MPN studies, which are dominated by PV and ET. Accordingly, the way forward seems to be to rewrite the paper such that it clearly states what it is already known, and what is novel, including the precise relationship between their findings and previously published (including for example precise LD statistics between identified and previously reported lead variants). The authors also need to remove all kinds of premature speculation, for example “our results indicate clonal expansion of JAK2 is facilitated by germline variation associated with longer telomere length” and “The findings have translation implications, highlighting the potential role for telomerase inhibitors as treatment in high-risk individuals”.

RESPONSE: Thank you for your thoughts and comments on our manuscript. We are in agreement that various results presented in our manuscript replicate past findings from MPN studies and have tried to clarify this point in several sections of our manuscript including the updated **Abstract**, **Results** and **Discussion**. Specifically, in our GWAS, we have included in the **Results** the precise LD statistics between the identified and previously reported lead variants as

well as cited references. Our findings are the first evidence suggesting MF is also associated with these MPN loci and they are not predominantly driven by PV or ET subtypes. We have further included a comment that the *TERT* variant was reported to be associated with blood CD34+ cell levels. Within the **Results** section: “Previously identified in MPN,²⁰ this intronic variant is located in *TERT*, which encodes telomerase, the reverse transcriptase that extends telomeric DNA repeats, and has been associated with CD34+ to CD45+ ratio.²⁵”

For the acquisition of *JAK2*^{V617F} mutations, we have cited the relevant literature mentioned by Reviewer 1. We updated the **Results** and **Discussion** to clarify that this association has been previously reported. Within the **Results** section: “Furthermore, when examining phase information, we observed a strong *cis* relationship between the germline risk haplotype and *JAK2*^{V617F} mutations acquired on the same risk haplotype (binomial $P=1.23 \times 10^{-26}$) (**Supplemental Table 5**), as previously observed in MPN patients.^{12–14}” Within the **Discussion** section: “We observed that individuals with the germline *JAK2* risk haplotype were predisposed to acquiring a somatic *JAK2*^{V617F} mutation in *cis*, as previously reported,^{12–14} and mCAs lead to preferential over-representation of this risk haplotype containing the *JAK2*^{V617F} mutation.⁴⁴”

For the telomere length findings, we compared our germline telomere length findings to the manuscript results (PMID33057200) mentioned by the reviewer. Within the **Results** section: “A marginally significant genome-wide genetic correlation was observed between telomere length and MF (LDSC $r=0.23$, s.e.m= 0.11, $P=0.038$), similar in magnitude to what was reported for telomere length and MPN (LDSC $r=0.19$, s.e.m= 0.09, $P=0.037$).²⁰”

Our study provides notable observations. First, our study is distinctive by performing integrated analysis of a large cohort (N=933) of clinically well annotated MF cases that included multiple layers of genomic data including genome-wide genotyping for germline risk analysis, estimating genetically predicted telomere length and detection of chromosomal alteration (mCAs), regional sequencing, and measured telomere length. Using this approach, we have demonstrated that the *JAK2* germline risk haplotype predisposes to the somatic acquisition of *cis* *JAK2*^{V617F} mutations, and mCAs acquired independently over the same region leading to preferential over-representation of this risk haplotype containing the *JAK2*^{V617F} mutation. This possibly predisposes to increased clonal expansion and associated telomere length shortening. Public posting of this MF germline array data in dbGaP promotes new opportunities for discovery and accelerates genetic research of MF susceptibility.

Second, our GWAS of MF replicated germline susceptibility loci identified in past MPN studies, confirmed associations with the MF subtype of MPNs, and identified two novel signals at 6p21.32 (*HLA-DRB9*) and 17p13.1 (*TP53*). These new loci reach the accepted multiple testing genome-wide significance level of $P < 5 \times 10^{-8}$, a level of statistical support that is commonly replicated in independent samples. The novel MF loci include biologically relevant candidate genes related to immunity and tumor suppression, both of which are potentially relevant to MF etiology. We acknowledge no independent replication of these new loci and based on Reviewer 1’s comment, we have added additional text about the need for further replication. Within the **Results** section: “Future studies are warranted to validate these two MF germline susceptibility loci.”

Third, our study represents the first large, comprehensive genome-wide assessment of mCAs in MF patients. We found highly elevated rates of mCAs in MF cases relative to population-based controls. We replicated prior observations that mCAs preferentially expand the *JAK2* MF risk haplotype and also identified a novel enrichment of mCAs at other MF GWAS susceptibility loci. Based on comments made by Reviewer 3, we further investigated mCA enrichment over the other MPN driver mutation positions (see new **Supplemental Table 10** below) and observed enrichment for mCAs at these common MPN driver mutations as well, suggesting mCAs may also selectively expand other non-*JAK2* MPN driver mutations.

Supplemental Table 10. mCA enrichment over myeloproliferative neoplasm driver mutation position^a

Region	Gene	Position (hg38)	MF mCAs, N(%)	UKBB mCAs, N(%)	p-value ^b
1p34.2	MPL	43337818-43354466	31 (3.65)	60 (0.141)	6.50×10^{-33}
19p13.13	CALR	12938609-12944489	25 (2.94)	28 (0.066)	1.36×10^{-32}

^aMyelofibrosis cases (N= 850) age (within 5-years) and sex-matched to UK Biobank cancer-free controls (N= 42,500)

^bBinomial test

mCA= mosaic chromosomal alteration

MF= Myelofibrosis

UKBB= UK Biobank

Finally, our study provides new insights into the importance of inherited telomere length as a likely risk factor for MF susceptibility and details the impact of *JAK2*-related clonal expansion on subsequent TL shortening. We used polygenic risk score (PRS) of recently published germline TL variants to demonstrate an association of longer inherited TL to MF risk. We found that the PRS was not only associated with increased MF risk but was further associated with both the presence of *JAK2*^{V617F} mutations as well as mCAs. These analyses suggest longer telomere length may afford cells with *JAK2*^{V617F} mutations or mCAs the ability to clonally expand to detectable clonal fractions, after acquiring these mutations. As reported previously and replicated in our MF study, measured rTL was not associated with *JAK2*^{V617F} mutation presence, but was inversely associated with *JAK2*^{V617F} clonal fraction (PMID: 23542632). Our results suggest mCAs are a predominant driver of this association as individuals with mCAs had the highest level of telomere length attrition, suggesting this group of MF patients may be at particularly high risk for progression and could represent a subset of MF cases more likely to benefit from telomerase inhibitors; although future studies are needed to test this hypothesis.

Reviewer #2 (Remarks to the Author): Expert in MF and MPN molecular genetics and genomics

The authors provide a comprehensive genomic analysis on very a large number of myelofibrosis patients compared with an even 5 times larger number of genetically-matched, cancer-free individuals serving as controls. The main methods comprise i) SNP genotyping to identify germline susceptibility loci, ii) calling for mosaic chromosomal alterations to determine copy number state, cellular fraction, and chromosomal region of the respective events, and iii) qPCR-based measurement of the relative telomere leukocyte length for association of identified SNPs with myelofibrosis risk and proliferation. The findings are in line with previous data on this topic and extend our understanding of the etiology of myelofibrosis due to new insights how germline variants and somatic alterations interact regarding dysregulation of JAK2 activity and thus clonal expansion in myelofibrosis patients.

In addition to known susceptibility loci in JAK2, TERT, TET2, and IFT80, the authors identified two novel germline loci in the genes HLA-DRB9 and TP53 predisposing for myelofibrosis acquisition. Furthermore, the autosomal location of both recurrent CNLOH events (mainly on chromosome 9p) and copy-number alterations (mainly recurrent loss events on chromosome 20q and 13q) is valid compared with former SNP array studies in myeloproliferative neoplasm patients. Including their data on telomere length, the authors finally provide a model in which the germline JAK2 risk haplotype predisposes to somatic acquisition of JAK2 V617F, subsequently causing clonal expansion and cell proliferation, and finally leading to accelerated telomere length shortening in myelofibrosis patients. These data and the proposed model justify the title of the manuscript. In summary, the presented manuscript is methodologically sound, very elaborate, thoughtfully written, and provides significant novelty to the field as well as for the readership of Nature Communications.

RESPONSE: Thank you for your review of our manuscript. We are glad you found the manuscript methodologically sound and novel. Please find our responses to your remarks below.

Minor remarks:

- The study is actually based on data from 827 myelofibrosis patients after quality control and matching with cancer-free individuals, not on 937 cases as stated in the Abstract.

Thank you for pointing this out. We have reworded the sentence in the Abstract to be “up to 933 myelofibrosis cases” as this is the largest N used for any analysis.

- The paper is in general a bit tough to read due to the high number of very particular abbreviations that are used. Please check for readability.

We have removed superfluous abbreviations where appropriate. We have also included an **Abbreviations** section at the end of the manuscript which we hope increases readability.

- Dynamic International Prognostic Scoring System = DIPSS (not DIPPS)

Thank you for drawing our attention to this typo. We have updated ‘DIPSS’ throughout the text and within all Figures and Tables.

Reviewer #3 (Remarks to the Author): Expert in MF and MPN genomics

In the present work authors perform an integrated genomic analysis on a wide cohort of primary and secondary myelofibrosis (PMF and SMF) patients in order to characterize the heritable genetic component of these neoplasms. As a result, authors identify six novel germline susceptibility loci, and among them two variants emerged, the first in 9p24.1 locus within an intron of the frequently mutated *JAK2* gene, and the second located within an intron of *TERT* gene. Authors show a correlation between the previously described *JAK2* 46/1 risk haplotype and the occurrence of *JAK2*V617F mutation or mosaic chromosomal alterations involving 9p24.1. According to their hypothesis, the presence of both germline variants, together with the acquisition of *JAK2*V617F mutation and somatic mosaic chromosomal alterations, contributes to the increased *JAK2* activity and clonal expansion as demonstrated by the reduction of telomere length. The presented results are significant and original, but I have some concerns about data interpretation.

RESPONSE: Thank you for your review of our manuscript. Please find our detailed responses to your comments below.

Major points

1. Myelofibrosis (MF) belongs to chronic Philadelphia-Negative Myeloproliferative Neoplasms (MPNs); it is generally assumed that the malignant transformation involves the myeloid lineage while lymphoid cells do not belong to the neoplastic clone, indeed in most cases these cells do not harbor the driver mutation observed in patients. Authors performed their analysis by studying whole blood specimens or PBMNCs therefore including in their analysis both neoplastic myeloid cells belonging to the expanded malignant clone (granulocytes, monocytes and circulating CD34+ cells) and lymphoid cells probably representing a non-malignant cell fraction. Does the presence of both clonal and non-clonal cells in the analyzed sample might influence analysis results? Can the variable frequency of neoplastic cells in the analyzed sample influence analysis results? In such a complex and variable scenario how were germline variants distinguished from somatic ones?

Thank you for the thoughtful questions. As you mentioned, our analyses were performed using DNA derived from either whole blood or PBMCs from our myelofibrosis patients. The mixture of mutated myeloid cells and normal lymphoid cells will not affect the germline genotyping data as the clustering algorithms we use in the Illumina germline genotype calling are tuned to call genotyped probes that follow Mendelian proportions for inherited homozygous and heterozygous variants. In addition, any somatic point mutations that arise in the MF cases would unlikely occur at the same position as a genotyping probe as the arrays only target 500K markers (0.02%) across the 3B base pair genome. To further confirm that neoplastic expansion of myeloid MF clones did not alter germline analysis results, we performed sensitivity analyses at the *JAK2* locus by removing any individual with a *JAK2* mCAs and the signal remained genome-wide significant ($P=1.99 \times 10^{-10}$).

For our mCA calling, we utilized the same genotype data used for our GWAS. When calling mCAs, we scan for large, contiguous genomic stretches with deviations in signal intensity and

allelic imbalances of heterozygous variants followed by QC steps that plots and visually inspects each potential mCA event. Only events that pass manual review were included in our analyses. As suspected, having a mixture of both myeloid and lymphoid cells does affect the estimation of clonal fraction of mCA events; biasing the estimated cellular fractions to lower levels based on the normal lymphoid cells in the evaluated DNA. As mCA detection limits are around 3-5% dependent on size and genomic location, it is also possible we miss low cell fraction mCA events present in myeloid cells due to the dilution from lymphoid cells. Most detected mCAs in our study were at high cell fraction (median= 36%), and therefore, we expect that influences of dilution from normal lymphoid cells had a minimal impact on detection of events in our study.

For our *JAK2*^{V617F} mutation calling using PacBio sequencing, we sequenced at a high depth (median coverage= 6,872X) so were unlikely to miss detection of low cell fraction *JAK2*^{V617F} mutations. We do however expect the *JAK2*^{V617F} cell fractions to have similar biasing of estimated cell fractions to lower values. As the same DNA samples were used for genotyping and sequencing, any dilution of the cell fraction will be controlled for when doing comparisons of cell fraction in the sequencing vs. genotyping data. Within the **Results** section we have added: “The estimated *JAK2*^{V617F} allelic fractions and mCA cellular fractions may be lower than the true somatic fraction in the actual diseased myeloid cells because we used whole blood DNA from most of the patients. However, this would not affect the ratio of *JAK2*^{V617F} allelic fraction to mCA cellular fraction.”

2. The present study is focused on MF that can be either primary or secondary, when it develops in patients who have a previous diagnosis of polycythemia vera (PV) or essential thrombocythemia (ET). Even if these two conditions (PMF and SMF) have been considered very similar for a long time, recent studies highlighted several differences in both clinical features and molecular characteristics. Related to the clinics, specific prognostic models were developed for risk prediction in SMF patients, in particular MYSEC-PM (Myelofibrosis Secondary to PV and ET – Prognostic Model). Taking this into account, it would be important to include, in Supplemental Table 2, the frequency of driver mutations (affecting *JAK2*, *MPL* and *CALR*) harbored by patients included in the study. Moreover, it is crucial to specify the frequency of Post-PV MF and Post-ET MF and not only the frequency of SMF. Finally, I warmly suggest to consider MYSEC-PM prognostic classification for SMF patients.

Thank you for these comments. We do not have complete clinical information on driver mutations (*MPL* and *CALR*) in all MF cases. Here, we performed targeted sequencing of the *JAK2* region to call *JAK2* mutations to follow-up on the *JAK2* MF susceptibility region from the GWAS analysis. We have an on-going sequencing project in this MF patient population where we will be able to evaluate those mutations in the future.

We have added the frequency of both Post-PV MF and Post-ET MF to **Supplemental Table 1**. This information is available for each of our analytical samples. Additionally, as detailed below in subsequent responses, we have performed new analyses which stratify SMF into Post-PV MF and Post-ET MF.

Thank you for your suggestion to use MYSEC-PM prognostic classification within SMF patients. We will use this classification in our future prognostic study after we complete the sequencing.

3. Related to the type of SMF, in Supplementary Table 2 would it be possible to perform the analysis by distinguishing between PostPV-MF and PostET-MF?

Based on your comment, we have performed two new GWAS stratified by Post-PV MF and Post-ET MF. Overall, there are 258 secondary MF cases in our GWAS. Stratified by type of SMF:

- N= 119 with Post-PV MF
- N= 139 with Post-ET MF

Below, please find new **Supplemental Table 3** and new **Supplemental Figure 3** which detail these SMF stratified GWAS results. Across each loci identified in our main GWAS study, all point estimates remain in the same direction in both the post-PV MF and post-ET MF analyses. Interestingly, the *JAK2* loci still reaches genome-wide significance within the post-PV MF analysis, which is consistent with prior reports of higher *JAK2* involvement in patients with PV. We have added these new GWAS findings to the **Results** section and have included the new **Supplemental Table 3** and new **Supplemental Figure 3** in the manuscript supplement.

Supplemental Table 3. Magnitude and strength of association for myelofibrosis susceptibility loci stratified by secondary myelofibrosis type

Region	Top SNP	Nearby gene	Position (hg38)	Ref	Risk	Post-Polycythemia Vera Myelofibrosis ^a		Post-Essential Thrombocythemia Myelofibrosis ^b	
						OR (95% CI)	p-value	OR (95% CI)	p-value
3q25.33	rs201009932	IFT80	160368930	T	TA	5.12 (1.70-15.42)	3.70×10 ⁻³	12.72 (4.01-40.28)	1.54×10 ⁻⁵
4q24	rs1548483	TET2	104828738	C	T	4.00 (1.83-8.73)	5.04×10 ⁻⁴	1.52 (0.82-2.82)	1.82×10 ⁻¹
5p15.33	rs7705526	TERT	1285859	C	A	2.13 (1.59-2.84)	2.87×10 ⁻⁷	1.73 (1.32-2.26)	7.52×10 ⁻⁵
6p21.32	rs28442287	HLA-DQB1-AS1	32658230	T	C	2.07 (1.32-3.26)	1.67×10 ⁻³	1.45 (0.97-2.18)	7.21×10 ⁻²
9p24.1	rs7851556	JAK2	5022807	C	T	5.31 (4.02-7.00)	4.06×10 ⁻³²	1.82 (1.36-2.42)	4.51×10 ⁻⁵
17p13.1	rs78378222	TP53	7668434	T	G	19.89 (6.21-63.72)	4.82×10 ⁻⁷	3.20 (1.22-8.38)	1.78×10 ⁻²

^aAnalysis performed in 119 cases and 595 controls

^bAnalysis performed in 139 cases and 695 controls

Ref= reference allele

Risk= risk allele

OR (95% CI)= myelofibrosis association odds ratio and 95% confidence interval adjusted for significant ancestry principal components

Supplemental Figure 3. Stacked Manhattan plots from the genome-wide association study stratified by post-polycythemia vera myelofibrosis (119 cases, 595 controls; Top plot) and post-essential thrombocythemia myelofibrosis (139 cases, 695 controls; Bottom plot). The association $-\log_{10}$ P-values are plotted for each tested genetic variant on the y-axis and chromosomal position on the x-axis. The nearest gene for each identified locus is labeled. The red line indicates the genome wide significance threshold (5×10^{-8}).

4. In paragraph “JAK2 germline risk haplotype confers elevated risk of *cis* JAK2V617F mutations” authors evaluate the association between a previously described JAK2 germline risk haplotype (JAK2 46/1) and the presence of JAK2V617F mutation. The authors report that the frequency of JAK2V617F in the study cohort is about 60% and is more frequent in SMF patients compared to PMF. This is obviously expected because it is already known that almost the totality of PV patients (therefore PostPV-MF patients) harbor JAK2V617F driver mutation. Which is the frequency of the JAK2 46/1 risk haplotype in the study cohort? Since the frequency of JAK2V617F mutation is greater in PostPV-MF patients compared to PMF and PostET-MF ones, would it be possible to evaluate the correlation between the presence of this mutation and the germline risk haplotype separately for each disease?

We have added the 46/1 haplotype frequency within our MF cases to the paragraph in question. We provide this for the overall patient population as well as stratified by PMF, post-PV MF, and post-ET MF.

Within the **Results** section: “... with 634 (68.61%) individuals carrying the *JAK2* 46/1 germline risk haplotype, and post-polycythemia vera MF (88.15%) having a higher frequency than both primary MF (65.98%) and post-essential thrombocythemia MF (66.45%).”

We have also performed new analyses that stratify the association between the *JAK2*^{V617F} mutation and germline haplotype within PMF, post-PV MF, and post-ET MF. Please see the tables below, for your reference, that detail these stratified results. We see that the overall *cis* relationship is still observed when stratified by each type of MF.

Within the **Results** section we now provide these new results: “These results were consistently observed when stratified by type of MF: primary MF (binomial $P= 1.86 \times 10^{-13}$), post-polycythemia vera MF (binomial $P= 4.88 \times 10^{-10}$), and post-essential thrombocythemia MF (binomial $P= 1.90 \times 10^{-4}$).”

Germline haplotypes by *JAK2*^{V617F} mutation status within Primary MF

	JAK2 ^{V617F} mutation status		Total	p-value ^a
	Mutation	No Mutation		
N	365	903	1,268	
Germline Haplotype				
GGC	216 (59.18)	268 (29.68)	484	1.86×10 ⁻¹³
TCT	146 (40.00)	628 (69.55)	774	2.93×10 ⁻¹⁰
GCC	3 (0.82)	3 (0.33)	6	0.3640
TGC	0	2 (0.22)	2	1
CCT	0	0	0	-
GCT	0	1 (0.11)	1	1
TCC	0	1 (0.11)	1	1

^aBinomial test

GGC= Germline risk haplotype

N= Number of germline haplotypes

Germline haplotypes by *JAK2*^{V617F} mutation status within post-PV MF

	JAK2 ^{V617F} mutation status		Total	p-value ^a
	Mutation	No Mutation		
N	131	139	270	
Germline Haplotype				
GGC	111 (84.73)	40 (28.78)	151	4.88×10 ⁻¹⁰
TCT	18 (13.74)	96 (69.06)	114	2.91×10 ⁻¹³
GCC	1 (0.76)	1 (0.72)	2	1
TGC	1 (0.76)	2 (1.44)	3	1
CCT	0	0	0	-
GCT	0	0	0	-
TCC	0	0	0	-

^aBinomial test

GGC= Germline risk haplotype

N= Number of germline haplotypes

Germline haplotypes by *JAK2*^{V617F} mutation status within **post-ET MF**

	JAK2 ^{V617F} mutation status		Total	p-value ^a
	Mutation	No Mutation		
N	66	244	310	
Germline Haplotype				
GGC	43 (65.15)	76 (31.15)	119	1.90×10 ⁻⁴
TCT	23 (34.85)	163 (66.80)	186	2.22×10 ⁻³
GCC	0	2 (0.82)	2	1
TGC	0	2 (0.82)	2	1
CCT	0	1 (0.41)	1	1
GCT	0	0	0	-
TCC	0	0	0	-

^aBinomial test

GGC= Germline risk haplotype

N= Number of germline haplotypes

5. The authors demonstrate that *JAK2*V617F is frequently observed in cis with the *JAK2* 46/1 risk haplotype. Please can the authors provide any observation about the co-occurrence of *JAK2*V617F mutation and the rs7851556 variant? Patients carrying the rs7851556 are more likely to have *JAK2*V617F mutation?

The reviewer's assumption is correct here. Our identified lead GWAS SNP (rs7851556) is in very high LD with the 46/1 risk haplotype. Below please find **Supplemental Table 4** which details these new results.

We have also added the following to the **Results** section: "MF cases carrying the risk allele (T) of rs7851556 (our lead GWAS SNP) were more likely to acquire a somatic *JAK2*^{V617F} mutation (P= 9.41×10⁻¹⁴) (**Supplemental Table 4**)."

Supplemental Table 4. *JAK2*^{V617F} mutation status by *JAK2* genotype (rs7851556) status

	rs7851556 Genotype			p-value ^a
	CC	TC	TT	
N	295	368	261	
JAK2 ^{V617F} mutation status				9.41×10 ⁻¹⁴
Mutation	150 (50.85)	202 (54.89)	210 (80.46)	
No Mutation	145 (49.15)	166 (45.11)	51 (19.54)	

^aChi-square test

6. JAK2V617F was the first driver mutation discovered in 2005 in MPN patients. It is the most frequent driver mutation affecting the vast majority (95%) of PV patients and almost 60% of PMF and ET ones. In the present study authors report that the interaction between germline and somatic variants in JAK2 contributes to the clonal expansion observed in MF patients. Can the authors discuss the possible involvement of the same mechanisms in other MPNs (PV and ET)? Can it be generalized that the presence of the risk haplotype rs7851556 is associated with an increased risk of acquiring a JAK2V617F mutation rather than the development of myelofibrosis?

As shown in our response to your above comment, we consistently observed the *cis* relationship between the 46/1 germline risk haplotype and the *JAK2*^{V617F} mutation for each type of MF (PMF, Post-PV MF, and Post-ET MF). Also, within our (new) stratified GWAS, we show that rs7851556 is associated with increased risk of PMF, Post-PV MF, and Post-ET MF, although the results for Post-ET MF do not reach genome-wide significance, probably due to the smaller sample size.

As rs7851556 is a strong LD tag for the 46/1 haplotype, we hypothesize that rs7851556 is not a specific risk factor for MF, but rather MPN as a whole, most likely due to the increased risk of acquiring a *JAK2*^{V617F} mutation, as shown previously (PMID 19287382, PMID19287384, and PMID 19287385).

7. Authors says that the presence of JAK2 risk haplotype, JAK2V617F mutation and 9p24.1 mosaic chromosomal alterations confer increased JAK2 activity. Can the authors provide any evidence or discussion on how germline haplotype might be able to influence protein activity, since the novel identified risk variant and those previously described are intronic variants that are not located in promoter region?

We agree with the reviewer that the identified germline risk variant (rs7851556) is an intronic variant, thus it is likely not a functional variant. Our eQTL and TWAS results indicate that the germline risk haplotype is associated with elevated JAK2 expression. It is likely that a variant in high LD with our identified risk variant is functional.

There is a 50+ Kb LD block within the *JAK2* region with our identified risk variant (plot given below). Using LDproxy (<https://ldlink.nci.nih.gov/?tab=ldproxy>), variants in LD with our risk variant can be evaluated for predicted functionality. There are several variants in high LD with the lead GWAS variant that demonstrate functional potential. Within the **Discussion** section we have further expanded on this point: “While we were able to connect germline susceptibility and somatic mutations in cross-sectional data from MF patients, future longitudinal assessment will be key in follow-up of our study. Likewise future studies characterizing germline functional variation are needed to better understand how the germline *JAK2* MF susceptibility locus leads to altered JAK2 expression and acquisition of *JAK2*^{V617F} mutations.”

8. Authors report that relative telomer length is inversely correlated with the increased JAK2V617F clonal fraction. As said by the authors, JAK2V617F mutated patients represent only about 60% of patients included in their study cohort. Can telomer length be correlated with clonal expansion even in cases where the MPN driver mutation is represented by CALR or MPL variants?

Thank you for this question. From our rTL analyses we observed:

1. An inverse association between measured rTL and presence of any autosomal mCA (OR= 0.14, 95% CI= 0.04-0.44, P= 7.39×10^{-4})
2. Individuals with mCAs in the 9p24.1 region had a stronger attenuation of rTL (OR= 0.04, 95% CI= 0.01-0.19, P= 2.60×10^{-5})

3. No association between measured rTL and *JAK2*^{V617F} mutation presence (OR= 0.84, 95% CI= 0.29-2.42, P= 0.74)
4. Inverse association between rTL and *JAK2*^{V617F} clonal fraction (β = -1.11, 95% CI= -1.31 - -0.91, P= 2.26×10^{-25})
5. After removing individuals with mCAs, rTL was no longer associated with *JAK2*^{V617F} clonal fraction (β = -0.27, 95% CI= -0.71 - 0.16, P= 0.22)

Together, these results suggest that it is not the *JAK2*^{V617F} mutation that is leading to clonal expansion, rather it is the mCAs, many of which selectively retain or duplicate *JAK2*^{V617F} mutations, that promote rapid clonal expansion of mutated clones resulting in significant reductions in telomere length.

We demonstrated in **Supplemental Table 7** that each GWAS susceptibility locus showed enrichment for mCAs compared to age and sex-matched cancer-free individuals in the UK Biobank, suggesting mCAs could clonally expand MF risk conferring alleles at susceptibility loci. Based on your suggestion, we further investigated mCA enrichment over the other MPN driver mutation positions (see new **Supplemental Table 10**). These results suggest that mCAs may also clonally expand other MPN driver mutations, but future studies with *MPN* and *CALR* sequencing data are needed to fully test this hypothesis.

We have added these findings to the **Results** and included the new **Supplemental Table 10** in the manuscript.

Supplemental Table 10. mCA enrichment over myeloproliferative neoplasm driver mutation position^a

Region	Gene	Position (hg38)	MF mCAs, N(%)	UKBB mCAs, N(%)	p-value ^b
1p34.2	MPL	43337818-43354466	31 (3.65)	60 (0.141)	6.50×10^{-33}
19p13.13	CALR	12938609-12944489	25 (2.94)	28 (0.066)	1.36×10^{-32}

^aMyelofibrosis cases (N= 850) age (within 5-years) and sex-matched to UK Biobank cancer-free controls (N= 42,500)

^bBinomial test

mCA= mosaic chromosomal alteration

MF= Myelofibrosis

UKBB= UK Biobank

Minor points

- Line 94: Authors say that primary myelofibrosis (PMF) patients “had an average time from diagnosis to transplant of 63.5 months (median= 25.1, IQR=9.0-88. 3).”, then, in line 96 it is reported that secondary myelofibrosis (SMF) “had longer average time from diagnosis to transplant (122.8 months vs. 36.8 months)”, it is not clear why the two mean values reported for primary myelofibrosis are different.

The first reported mean value was for the full cohort, and the subsequent mean values are for secondary and primary MF, respectively. We have updated the sentence, accordingly.

Within the **Results** section: “The average time from diagnosis to transplant in the full cohort was 63.5 months (median= 25.1, IQR=9.0-88. 3). Compared to primary MF cases, secondary MF cases were more likely to be female (54.45% vs. 36.28%) and had longer average time from diagnosis to transplant (122.8 months vs. 36.8 months).”

REVIEWERS' COMMENTS

Reviewer #3 (Remarks to the Author):

To the Authors: please find my new comments to your answers.

1.

Thank you to the Authors for their answers. Now, it is clear to me why genotyping analysis for the GWAS is not influenced by the presence of neoplastic mutated cells in analyzed samples. It is also clear and expected that in such a situation where a mixture of both neoplastic and normal cells is analyzed the observed frequency of JAK2V617F and mCAs is underestimated due to a dilution effect.

2.

The authors did not provide MYSEC-PM classification and the frequency of CALR and MPL mutated patients because analysis are still on-going. I believe that, to date, the identification of the driver mutation responsible for disease development is of primary importance to obtain the most significant results in studies involving myelofibrosis patients cohorts.

The authors included SMF classification distinguishing Post-PV MF from Post-ET MF as suggested.

3.

In the revised version of the paper the authors stratified SMF patients according to diagnosis of Post-PV MF or Post-ET MF. They performed new GWAS considering Post-PV MF and Post-ET MF separately and showed it in Supplemental Table 3 and Supplemental Figure 3. In these two cohorts only the identified JAK2 loci rs7851556 reached the statistical significance in only Post-PV cases. Even if the absence of significance was related by the authors to the low number of Post-ET MF cases I believe this should be said in discussion line 375 – 376.

4.

The authors answered my questions. They included the observed frequency of JAK2 46/1 risk haplotype in studied population stratifying patients according to diagnosis. Included this data in Results together with the observed co-occurrence between JAK2 46/1 risk haplotype and JAK2V617F mutation in patients stratified according to disease diagnosis.

Nevertheless, to my opinion, it is not clear what is to be considered original result in this paragraph, since as I told before, it is already known that the frequency of JAK2V617F mutation is higher in Post-PV MF (lines 174-177). Moreover, it is known that the presence of JAK2 46/1 haplotype predisposes to the development of JAK2 mutated MPN, according to papers cited by the authors, therefore it is not clear to me the novelty of results presented in this paragraph.

5.

The authors observed the co-occurrence of JAK2V617F mutation and the identified lead GWAS SNP rs7851556 showing that MF patients carrying the risk allele (T) of rs7851556 are more likely to acquire a JAK2V617F mutation. Result of this analysis is reported in Supplemental Table 4. Related to comment above, it is not clear why authors did not deepen the characterization of the identified lead SNP rs7851556 in favor of the already described JAK2 46/1 haplotype. Which is the frequency of rs7851556 risk variant in the study cohort? Would it be possible to identify a cis relationship between rs7851556 and JAK2V617F ?

6.

The authors answered the question. Related to this, I think that in discussion there is no clear reference to the identified lead variant rs7851556. I think that a reference to rs7851556 should be included in line 338 339, otherwise it seems to be referred only to previously described risk haplotype.

7.

The authors agree with me saying that rs7851556 germline risk variant is not likely to be a functional variant since it is located in an intronic region. Using LDproxy they identified many other variants in LD with the identified risk one. None of these variants have been demonstrated to be functionally relevant. This is one of the main limit of the study, since the title of the paper points toward an involvement of genetic variants in clone expansion determination without demonstrating a true functional effect of the described variant.

It has been demonstrated by authors that the presence of risk variant rs7851556 correlates with the presence of JAK2V617F mutation. It is known that JAK2V617F mutation correlates with increased JAK2 expression (PMID: 24740812). Would it be possible that the correlation between risk variant and gene expression is determined by the effect of JAK2V617F mutation itself?

Related to this, in discussion Line 343 – 345 authors says “Increased JAK2 activity conferred by the germline JAK2 risk haplotype, JAK2V617F activating mutation, and 9p24.1 mCAs promotes a cellular phenotype characterized by increased clonal expansion.” This represents an author assumption, indeed they observed increased expression of JAK2 correlating with the presence of rs7851556 risk variant, not an increased activity of JAK2. I believe authors should state clearly this is their leading hypothesis. As they say in following lines a functional demonstration on how these factors cooperate in promoting clonal expansion is missing. In line 369-370 authors say that flow-FISH might be useful to test their hypothesis, did they have any preliminary results that associate telomer length with clonal expansion and the presence of JAK2 mCAs in MF patients?

8.

The authors answered the question and identified mCAs affecting other MPN driver mutations positions that may contribute to clonal expansion in JAK2 negative patients. Did the authors observed any association with reduced rTL suggesting clonal expansion also in patients with mCAs affecting CALR and MPL genomic regions?

Minor Comments

- Line 173 remove "Also", JAK2V617F mutation is a driver mutation in MPN.
- Line 179 "mutation" should be "mutated"
- Line 575 change DIPPS with DIPSS

Author response

We thank the Reviewer for their time and continued review of our manuscript. Below, please find our responses to each of the comments, as well as where in the manuscript any revisions can be found. All changes have been tracked with **highlighted text**.

REVIEWERS' COMMENTS

Reviewer #3 (Remarks to the Author):

To the Authors: please find my new comments to your answers.

1. Thank you to the Authors for their answers. Now, it is clear to me why genotyping analysis for the GWAS is not influenced by the presence of neoplastic mutated cells in analyzed samples. It is also clear and expected that in such a situation where a mixture of both neoplastic and normal cells is analyzed the observed frequency of JAK2V617F and mCAs is underestimated due to a dilution effect.

We are glad the Reviewer found our response adequately explained the robustness of our genotyping analysis as well as how we estimate mosaic cell fractions.

2. The authors did not provide MYSEC-PM classification and the frequency of CALR and MPL mutated patients because analysis are still on-going. I believe that, to date, the identification of the driver mutation responsible for disease development is of primary importance to obtain the most significant results in studies involving myelofibrosis patients cohorts. The authors included SMF classification distinguishing Post-PV MF from Post-ET MF as suggested.

Thank you for your suggestion to use MYSEC-PM prognostic classification within SMF patients. We agree with the importance of identifying driver mutations and plan to use this classification in future studies when lengthy clinical data abstractions and sequencing efforts are complete.

3. In the revised version of the paper the authors stratified SMF patients according to diagnosis of Post-PV MF or Post-ET MF. They performed new GWAS considering Post-PV MF and Post-ET MF separately and showed it in Supplemental Table 3 and Supplemental Figure 3. In these two cohorts only the identified JAK2 loci rs7851556 reached the statistical significance in only Post-PV cases. Even if the absence of significance was related by the authors to the low number of Post-ET MF cases I believe this should be said in discussion line 375 – 376.

Based on your comment we have added an additional point to the Discussion about Post-PV MF and the JAK2 risk loci. Within the **Discussion**: “The large number of cases for this rare disease allowed for investigation of potential differences between primary and secondary MF, and secondary subtype. We noted no major differences in germline susceptibility by type of MF, and observed a significant signal at 9p24.1 (JAK2) in post-polycythemia vera MF.”

4. The authors answered my questions. They included the observed frequency of JAK2 46/1 risk haplotype in studied population stratifying patients according to diagnosis. Included this data in Results together with the observed co-occurrence between JAK2 46/1 risk haplotype and JAK2V617F mutation in patients stratified according to disease diagnosis. Nevertheless, to my opinion, it is not clear what is to be considered original result in this paragraph, since as I told before, it is already known that the frequency of JAK2V617F mutation is higher in Post-PV MF (lines 174-177). Moreover, it is known that the presence of JAK2 46/1 haplotype predisposes to the development of JAK2 mutated MPN, according to papers cited by the authors, therefore it is not clear to me the novelty of results presented in this paragraph.

We are glad our previous response adequately addressed your questions. While the cited literature has identified the *JAK2* 46/1 haplotype predisposes to the development of *JAK2*^{V617F}, we show this relationship is consistent within MF patients stratified by primary, post-PV, and post-ET MF patients. Additionally, our results support previous studies indicating a higher frequency of *JAK2*^{V617F} mutations in Post-PV MF

5. The authors observed the co-occurrence of JAK2V617F mutation and the identified lead GWAS SNP rs7851556 showing that MF patients carrying the risk allele (T) of rs7851556 are more likely to acquire a JAK2V617F mutation. Result of this analysis is reported in Supplemental Table 4. Related to comment above, it is not clear why authors did not deepen the characterization of the identified lead SNP rs7851556 in favor of the already described JAK2 46/1 haplotype. Which is the frequency of rs7851556 risk variant in the study cohort? Would it be possible to identify a cis relationship between rs7851556 and JAK2V617F ?

Thank you for your comment. The frequency of the rs7851556 risk variant (T) is 48%. Within Supplemental Table 4 we display *JAK2*^{V617F} mutation status by *JAK2* genotype (rs7851556) status, with all the genotype counts. The distance between our lead GWAS variant (rs7851556) and the *JAK2*^{V617F} mutation is approximately 50Kb (chr9:5022807- 5073770) in length. Repetitive regions in this area lead to several challenges with sequencing. To avert these obstacles, we sequenced a 7Kb region around the *JAK2*^{V617F} mutation which included variants in the 46/1 haplotype. We demonstrated a *cis* relationship between the 46/1 haplotype and *JAK2*^{V617F} mutation and as these variants in the 46/1 risk haplotype are in high LD with our lead GWAS SNP ($R^2 > 0.93$, see below plot), a *cis* relationship between rs7851556 and the *JAK2*^{V617F} mutation is established.

Within the above plot, our lead GWAS variant (rs7851556) is colored blue. The star represents the $JAK2^{V617F}$ mutation, and the red box is the 7Kb region we sequenced.

6. The authors answered the question. Related to this, I think that in discussion there is no clear reference to the identified lead variant rs7851556. I think that a reference to rs7851556 should be included in line 338 339, otherwise it seems to be referred only to previously described risk haplotype.

Thank you for your comment. We have added a reference to the identified lead GWAS variant within the section mentioned. Within the Discussion: “We observed that individuals with the germline $JAK2$ risk haplotype tagged by rs7851556 were predisposed to acquiring a somatic $JAK2^{V617F}$ mutation in *cis*, as previously reported...”

7. The authors agree with me saying that rs7851556 germline risk variant is not likely to be a functional variant since it is located in an intronic region. Using LDproxy they identified many other variants in LD with the identified risk one. None of these variants have been demonstrated to be functionally relevant. This is one of the main limit of the study, since the title of the paper points toward an involvement of genetic variants in clone expansion determination without demonstrating a true functional effect of the described variant.

It has been demonstrated by authors that the presence of risk variant rs7851556 correlates with the presence of $JAK2^{V617F}$ mutation. It is known that $JAK2^{V617F}$ mutation correlates with increased $JAK2$ expression (PMID: 24740812). Would it be possible that the correlation between risk variant and gene expression is determined by the effect of $JAK2^{V617F}$ mutation itself?

Related to this, in discussion Line 343 – 345 authors says “Increased $JAK2$ activity conferred by the germline $JAK2$ risk haplotype, $JAK2^{V617F}$ activating mutation, and 9p24.1 mCAs promotes a cellular phenotype characterized by increased clonal expansion.” This represents an author assumption, indeed they observed increased expression of $JAK2$ correlating with the presence of rs7851556 risk variant, not an increased activity of $JAK2$. I believe authors should state clearly this is their leading hypothesis. As they say in following lines a functional demonstration on how

these factors cooperate in promoting clonal expansion is missing. In line 369-370 authors say that flow-FISH might be useful to test their hypothesis, did they have any preliminary results that associate telomere length with clonal expansion and the presence of JAK2 mCAs in MF patients?

Our study deeply characterized germline and somatic profiles (including *JAK2*^{V617F} mutations and mCAs) in a large population of MF cases. We localized germline variation at 9p24.1 to the tagging marker rs7851556 and provided evidence for how this germline variation influences somatic profiles in MF cases as well as *JAK2* expression levels in normal circulating leukocytes (GTEx). As GWAS identifies regions of germline variation important for MF risk, we make no claims of function for the rs7851556 variant as it likely tags other germline variation with functional relevance. We agree with the Reviewer that rs7851556 is likely not functional and recommended in the Discussion section further studies to identify functionally relevant variation in the region.

rs7851556 could have effects through promoting *JAK2* expression (as observed in our eQTL analyses in normal whole blood) as well as through leading to susceptibility of the *JAK2*^{V617F} and mCA changes that have downstream impacts on *JAK2* activity and expression (PMID: 24740812), but extensive functional work would be needed and teasing out impacts of rs7851556 vs *JAK2*^{V617F} is challenging due to high correlation. As we do not have laboratory capacity to assess the functional mechanism by which the risk variant alters *JAK2* expression, our analyses and inferences lean heavily on public resources and prior published reports.

For the Discussion section sentence in question, we agree that this is our hypothesis. To clarify, we have modified the text as follows: “Altered *JAK2* activity conferred by the germline *JAK2* risk haplotype, *JAK2*^{V617F} activating mutation, and 9p24.1 mCAs could lead to a cellular phenotype characterized by increased clonal expansion”

We do not have flow-FISH data to present with regards to telomere length. Functional work is beyond the intended scope of our study’s analytical aims; however, existing published reports along with evidence from public data lend strong support for our proposed etiologic framework

8. The authors answered the question and identified mCAs affecting other MPN driver mutations positions that may contribute to clonal expansion in *JAK2* negative patients. Did the authors observed any association with reduced rTL suggesting clonal expansion also in patients with mCAs affecting *CALR* and *MPL* genomic regions?

In our analyses, we observed a strong relationship in which increasing mCA cellular fraction was associated with a substantial decrease in measured rTL ($\beta = -0.57$, 95% CI = $-0.74 - -0.39$, $P = 4.76 \times 10^{-10}$). These analyses were conducted for any mCA. We did not perform mCA cellular fraction analyses subset to mCAs spanning *CALR* or *MPL* mutations due to the limited sample size (N=25 and 31, respectively), but our overall analyses suggest that the inverse association between rTL and cellular fraction is consistent regardless of mCA position.

Minor Comments

- Line 173 remove “Also”, JAK2V617F mutation is a driver mutation in MPN.
Removed.

- Line 179 “mutation” should be “mutated”
Updated.

- Line 575 change DIPPS with DIPSS
Updated.